# The Role of VEGF Receptors as Molecular Target in Nuclear Medicine for Cancer Diagnosis and Combination Therapy

**DOI:** 10.3390/cancers13051072

**Published:** 2021-03-03

**Authors:** Katarzyna Masłowska, Paweł Krzysztof Halik, Dagmara Tymecka, Aleksandra Misicka, Ewa Gniazdowska

**Affiliations:** 1Centre of Radiochemistry and Nuclear Chemistry, Institute of Nuclear Chemistry and Technology, Dorodna 16, 03-195 Warsaw, Poland; e.gniazdowska@ichtj.waw.pl; 2Faculty of Chemistry, University of Warsaw, Pasteura 1, 02-093 Warsaw, Poland; dulok@chem.uw.edu.pl (D.T.); misicka@chem.uw.edu.pl (A.M.)

**Keywords:** VEGF, VEGF receptors, radiopharmaceuticals, anti-angiogenic therapy

## Abstract

**Simple Summary:**

The rapid development of diagnostic and therapeutic methods of the cancer treatment causes that these diseases are becoming better known and the fight against them is more and more effective. Substantial contribution in this development has nuclear medicine that enables very early cancer diagnosis and early start of the so-called targeted therapy. This therapeutic concept compared to the currently used chemotherapy, causes much fewer undesirable side effects, due to targeting a specific lesion in the body. This review article discusses the possible applications of radionuclide-labelled tracers (peptides, antibodies or synthetic organic molecules) that can visualise cancer cells through pathological blood vessel system in close tumour microenvironment. Hence, at a very early step of oncological disease, targeted therapy can involve in tumour formation and growth.

**Abstract:**

One approach to anticancer treatment is targeted anti-angiogenic therapy (AAT) based on prevention of blood vessel formation around the developing cancer cells. It is known that vascular endothelial growth factor (VEGF) and vascular endothelial growth factor receptors (VEGFRs) play a pivotal role in angiogenesis process; hence, application of angiogenesis inhibitors can be an effective approach in anticancer combination therapeutic strategies. Currently, several types of molecules have been utilised in targeted VEGF/VEGFR anticancer therapy, including human VEGF ligands themselves and their derivatives, anti-VEGF or anti-VEGFR monoclonal antibodies, VEGF binding peptides and small molecular inhibitors of VEGFR tyrosine kinases. These molecules labelled with diagnostic or therapeutic radionuclides can become, respectively, diagnostic or therapeutic receptor radiopharmaceuticals. In targeted anti-angiogenic therapy, diagnostic radioagents play a unique role, allowing the determination of the emerging tumour, to monitor the course of treatment, to predict the treatment outcomes and, first of all, to refer patients for AAT. This review provides an overview of design, synthesis and study of radiolabelled VEGF/VEGFR targeting and imaging agents to date. Additionally, we will briefly discuss their physicochemical properties and possible application in combination targeted radionuclide tumour therapy.

## 1. Introduction

The process of new blood vessel creation in cancer formation and growth, as well as the influencing factors, has been at the forefront of cancer research over the last few decades [1,2,3]. It is now known that vascular endothelial growth factor (VEGF) and vascular endothelial growth factor receptors (VEGFRs) play a pivotal role in angiogenesis process [3,4,5,6,7]. Nowadays, the use of inhibitors of angiogenesis promoting factors is a powerful tool in anticancer combination therapeutic strategies [3,5,8,9,10,11,12,13,14,15,16,17,18,19,20,21,22,23]. Several types of molecules have been used in targeted VEGF/VEGFR anticancer therapy: human VEGF ligands and their derivatives, anti-VEGF or anti-VEGFR monoclonal antibodies (mAb) (e.g., bevacizumab, ranibizumab) [19,24], VEGFR binding peptides and proteins [13] and small molecular inhibitors of receptor tyrosine kinases (RTKs) of VEGF receptors (e.g., sunitinib, sorafenib, vandetanib) [8,16,18,25,26,27,28]. Up to 20 RTK inhibitors have been approved for clinical use, in which certain inhibitors specifically target VEGFR, while others act as multi-kinase inhibitors [26,29,30]. Additionally, neuropilin 1 (NRP-1) a major co-receptor of VEGFR-2, as well as an independent receptor, is involved in the regulation of physiological and pathological angiogenic processes. For this reason, current research has focused on various neuropilin-targeting substances due to its possible application of anticancer therapy [31,32,33,34,35,36,37,38,39,40,41].

It is generally accepted that targeted anticancer drugs are the most effective in target overexpressing tumours; however, several clinical studies have shown that in different patients with the same tumours, the effects of these drugs are not always sufficiently efficient [42,43,44]. The ultimate goal of contemporary personalised oncology is to tailor the specific treatment protocols at the right time for individual patient population and to provide quantitative, low-invasive and accurate information on their responses to the therapy in the real time [15,16,20,45,46,47].

Modern methods of molecular imaging (SPECT, PET), defined as in vivo targeted visualisation for assessment of biological processes, provide an insight into physiological or pathological processes at the molecular level. They are considered as the basis for accurate clinical diagnosis, providing almost immediate feedback about the course of the treatment process. Receptor radiopharmaceuticals play the leading role in nuclear medicine, which is at the forefront in both cancer diagnosis and combination targeted therapy. These radiopharmaceuticals contain an appropriate (diagnostic or therapeutic) radionuclide and biovector (antibody, peptide or small organic molecule), that leads the radiopharmaceutical mostly to its specific receptors overexpressed on tumour cells. The use of biovectors in nuclear medicine enables the accumulation of diagnostic or therapeutic radionuclide mainly at the target site and achieve a high target to non-target ratio. As a consequence, that provides high-resolution imaging, or enables effective and specific annihilation of cancer cells.

Anti-angiogenic therapy (AAT) is a particular type of cancer treatment approach, which employs VEGFRs and its ligands as a reliable molecular target of tumour associated angiogenesis [3,15,16,20]. Angiogenesis is physiological vital process occurring during wound healing and embryonic development, that provides the formation of new capillaries splitting from created vessels over the vasculogenesis process. It serves as the key mechanism to overcome local metabolic stress and intracellular hypoxia in pathological conditions. It is known that, many tumours force neoangiogenesis as an essential growth and nutrition constituent and further to initiate metastasis. The pharmacologic inhibition of angiogenesis via VEGF pathway is considered an underestimated therapeutic support in prevention of cancer development and metastasis formation. Several studies have shown that anti-VEGF treatment, in association with chemotherapy or radiation therapy, results in greater anti-tumour effects compared to independent treatment [46].

The aim of this review is to collect and discuss data regarding the chemical and biological aspects of radiolabelled VEGF derivatives and VEGFRs’ ligands used in nuclear medicine for cancer diagnosis and combination targeted radionuclide tumour therapy. Specific consideration has been placed on the role of radiolabelled VEGF derivatives and VEGFRs’ ligands in AATs. 

## 2. VEGF/VEGFRs System in Angiogenesis and AAT

### 2.1. VEGF Glycoproteins

The first reports on VEGF appeared in 1980s, when it was recognised as vascular permeability factor [48], vasculotropin [49] and, as currently known, vascular endothelial growth factor [50], an endogenous effector of prominent pro-angiogenic action through direct activation of vascular endothelial cells. VEGF belongs to the mammalian peptide family consisting of constituents originating from different genes: VEGF-A, VEGF-B, VEGF-C, VEGF-D and PlGF (placenta growth factor), but also viral homolog VEGF-E [51] and VEGF-F of snake venom origin [52]. The common feature of these glycoproteins is the creation of dimeric forms through specific sequence of cysteines forming disulphide bridges between two monomers [53]. Each VEGF family protein occurs as a glycosylated peptide monomer; however, it has to homodimerise or heterodimerise to activate its biological function. 

VEGF-A (commonly called VEGF), is the most researched representative of the family and occurs in multiple isoforms (e.g., VEGF-A_121_, VEGF-A_145_, VEGF-A_165_, VEGF-A_183_, VEGF-A_189_ and VEGF-A_206_) due to an alternative splicing of mRNA obtained in the transcription process of the human gene *VEGFA* [54,55]. The *VEGFA* gene consists of eight exons that are highly conserved between species. In the first five constitutive exons are encoded the fundamental signal sequence, dimerisation cysteine fragment, specific VEGF receptors recognition domain, fragment employed in glycosylation and plasmin cleavage site, respectively. Furthermore, exons 6 and 7 encode an alternative heparine binding sequence and neuropilin binding domain, while last exon 8 encodes the unique VEGF domain. Alternative splicing results in variability of the primarily structure between isoforms, which affects their bioavailability and biological potency, mainly due to the isoform affinity to heparin sulphate and proteoglycan present on the extracellular surface competing with VEGF receptors [56]. Therefore, VEGF-A_121_ is freely diffusible and highly active isoform because it binds to neither neuropilins nor heparin sulphate, while VEGF-A_165_ and VEGF-A_189_ bind to both, resulting in expansion of their retention on the cellular surface or extracellular matrix. 

Althought VEGF-A is highly recognised as a critical angiogenic inductor, it shows broad pleiotropic action in mammals, namely,
(I)significant mitogenic effect on vascular endothelial cells [57], as well as anti-apoptotic impact on these cells [58];(II)increase of vascular permeability, resulting in increased serum peptides extravasation and local intra-tissue pressure [59];(III)induction of chemotaxis and activation of monocytes and haematopoietic stem cells [60,61,62];(IV)neurotrophic and neuroprotective action [63].

The production of VEGF-A glycoproteins occurs in the endothelium and vascular smooth muscle cells, but also in activated platelets, fibroblasts, lymphocytes and macrophages [64], where the production may be stimulated by numerous factors. This process is especially noticeable in tumour cells, that hyperexpress VEGF to stimulate the promotion of tumour growth neoangiogenesis [65]. The main initiator of the transcription of mRNA encoding VEGF-A is hypoxia state, especially noticeable in the necrotic and cancer cells [66]. This phenomenon is associated with the formation of hypoxia induced factor in these cells, which is called hypoxia inducible factor-1 (HIF-1) [67,68]. In contrast to hypoxia, HIF-1 cellular concentration is strictly regulated under physiological conditions. Other significant stimulating factors of VEGF-A cellular synthesis are cytokines (interleukin 1b, IL-1b and tumour necrosis factor alpha, TNF-α), several hormones and specific growth factors [69,70], activation of oncogenes RAS and SRC, mutation in suppressor genes p53 and von Hippel–Lindau (VHL) [70,71,72], as well as nitric oxide and oxygen radicals [73,74]. These factors are more or less known as indirect initiators of angiogenesis, acting on the synthesis of VEGF-A. 

The activity of other mammalian VEGF proteins is more specific than that of VEGF-A, however effects in site of action of all VEGF glycoproteins are more or less similar. VEGF-B has a relatively limited angiogenic action only towards ischemic myocardium, which is associated with VEGF-B level decrease [75]. More recently, it has been revealed that potent metabolic and antioxidative action of VEGF-B is possibly related to pro-angiogenic effects [76,77,78]. It contributes to the homeostasis of lipids in numerous tissues and the upregulation of brown adipose tissue, resulting in reduced risks of obesity and insulin resistance induced by diet rich in fat. Moreover, there are also reports of neuroprotective activity of exogenous VEGF-B_186_ isoform in the distal neuropathy and Parkinson’s disease models [79,80]. This effect is assumed to be induced directly on the motor neurons, similar to VEGF-A, not through their vascularity.

Some similarities to VEGF-B action exhibits placenta growth factor. PlGF is expressed dominantly by placental trophoblasts, but also during early embryonic development and to a lesser extent in a few adult organs such as heart, lungs, thyroid or skeletal muscles [81]. Contribution of PlGF in physiological angiogenesis in adults is negligible, however under pathological conditions such as ischemia, it prominently stimulates vascular endothelium proliferation and also differentiation and activation of the monocytes into the macrophages recognised as an angiogenic feedback stimulant [82]. Moreover, PlGF increases vessel permeability and inflammation in degenerations as rheumatoid arthritis and atherosclerosis promoting neoangiogenesis [83]. In addition, several types of tumour cell lines have the ability of PlGF expression, which favours the pro-angiogenic M2-phenotype tumour-associated macrophages [84].

VEGF-C is recognised as the fundamental promotor of proliferation and migration of the lymphatic system endothelium [85]. It also stimulates the cytokine-inducted migration and permeability of the vascular endothelial cells, although to a lesser extent than VEGF-A and independently of hypoxia stimulus. Similar in structure and function to VEGF-C, VEGF-D plays a secondary role in the physiological stimulation of human endothelium of vascular and lymphatic systems. Concomitantly, the high expression of both growth factors significantly promote and correlate with the metastasis through the lymphatic vessels in a variety of cancers [86,87,88].

### 2.2. VEGF Receptors and Their Co-Receptors

The site of action of all above growth factors are specific receptors for VEGF glycoproteins on the surface of target cells. There are three such receptors: VEGFR-1 (also known as FLT1, due to the same name of its gene), VEGFR-2 (known as KDR or FLK1, encoded by *KDR* gene) and VEGFR-3 (FLT4, encoded by *FLT4* gene). 

VEGFRs are classified as members of receptor tyrosine kinase superfamily due to their autophosphorylation ability induced by recognition of specific ligands. They are present in the form of homo- or heterodimers consisting of three functional fragments defined as extracellular part with seven Ig-like subunits, lipophilic single transmembrane domain and intracellular domain with distinctive tyrosine kinase activity. Individual VEGF proteins (and their isoforms) have different affinity towards each receptor. It is well known that VEGFR-1 binds VEGF-A, VEGF-B and PlGF, while VEGFR-2 binds VEGF-A as well as post-proteolytic VEGF-C and VEGF-D. Both VEGF-C and VEGF-D have affinity mainly towards VEGFR-3 [89] (Figure 1).

Interaction of growth factor with its receptor becomes much stronger with the participation of specific co-receptors that facilites the creation of the molecular complex ligand-receptor [90]. These co-receptors, known as neuropilins, occur as neuropilin 1 (NRP-1) that participates in VEGFR-1 or VEGFR-2 interactions with ligands and neuropilin 2 (NRP-2) mostly assigned to VEGFR-3 (Figure 1). Both types of neuropilins are expressed on endothelial cells and specific types of tumours [90,91]. NRP-1 binding differs between VEGF isoforms, so that VEGF-A_165_ and VEGF-A_189_ create stronger complexes with VEGFR-2 and NRP-1 than VEGF-A_121_, which is deprived of NRP-1 binding domain [90]. Nevertheless, direct interaction of VEGF-A_121_ with NRP-1 can regulate endothelial cell migration and sprouting independently of specific VEGF receptors [92].

The expression of VEGFR-1 occurs predominantly on endothelial cells of blood vessels, but also on monocytes and macrophages, placental trophoblasts as well as renal mesangial cells [93,94]. Similarly, VEGFR-2 occurs mostly on blood vessel endothelium, as well as platelets, haematopoietic and retinal stem cells. Both receptors are clearly expressed on cell surfaces of solid cancers and haematopoietic system neoplasms [95,96]. VEGFR-3 expression is specified only on endothelial cells of lymphatic system [97]. Therefore, a substantial share of VEGFR-1 and VEGFR-2 on vascular endothelium shows their significant contribution in angiogenesis, while VEGFR-3 and NRP-2 highly contribute in lymphangiogenesis [89,98]. 

For ligand binding receptors require at least the first three Ig-like domains, however, not all must participate in ligand binding. Simultaneously, if the ligand binds to neuropilin, then the third and fourth domains of the receptor will also attach to neuropilin. Moreover, besides ligand interaction, receptors also have to dimerise to be able to transduct signals intracellularly [99,100]. When both conditions are met, ligand can trigger the mutual autophosphorylation of the receptor intracellular tyrosine subunits and activation of specific signalling pathways inside the cell.

Different ligands can stimulate various biological effects through activated receptors, as well as activation of VEGFR-1 and VEGFR-2 by VEGF-A cause a different induction of intracellular signalling pathways [58,100]. Activation of VEGFR-2 leads to stimulation of the cell cycle, proliferation, migration, cell differentiation, angiogenesis, increased permeability of blood vessels but also inhibition of the apoptotic death and up-regulation of VEGF-A synthesis in endothelial cells [58,101]. On the contrary, VEGF-A can bind to VEGFR-1, activating its low-efficient tyrosine kinases, which has insignificant influence on endothelial cells [100,102]. Despite the high abundance of this receptor on endothelium, second receptor exerts even 10-fold higher density on endothelial cells [100,103]. Concomitantly, VEGF-A has about 10-fold lower affinity to VEGFR-2 compared to VEGFR-1. Hence, it is suspected that VEGFR-1 acts as concomitant decoy receptor and uptakes VEGF-A before it can bind to adjacent VEGFR-2, ergo VEGFR-1 plays an angiogenic-regulation role [82,103]. However, the same receptor interaction with PlGF promotes VEGF-A pool for endothelial angiogenic action through VEGFR-2 [82] and can regulate transphosphorylation of VEGFR-2 [104], thus amplifying angiogenesis through VEGFR-2. VEGFR-1 signalling can also regulate paracrine release in the vascular endothelial cells of other tissue endothelium growth factors inducing intestinal organogenesis and morphogenesis before vascular flow formation [105].

All VEGF isoforms that bind selectively to VEGFR-2 are capable to elicit receptor autophosphorylation, thus triggering the activation of numerous intracellular signalling pathways (Figure 2) [58,100,103,106,107]. Phosphorylated receptor subunits bind many adaptor molecules such as Shb (SH2 domain-containing adapter protein B), SOS (Son of sevenless proteins) or Grb-2 (Growth factor receptor-bound protein 2) that activate Ras GPTase. This last protein stimulates MAPK pathway responsible for endothelium proliferation. Simultaneously, phosphorylated intracellular VEGFR-2 domain activates phospholipase C-gamma (PLC-γ), which catalyses hydrolysis of phosphatidylinositol bisphosphate (PIP_2_) to inositol triphosphate (IP_3_) and diacylglycerol (DAG). IP_3_ triggers intracellular release of Ca^2+^ form endoplasmic reticulum, which employs calcium modulated protein calmodulin to stimulate cAMP phosphodiesterase, adenylate cyclase and site-specific endothelial NO synthase (eNOS) and consequently increase NO-driven vasodilation and vascular permeability. However, DAG activates calcium-dependent protein kinase C (PKC), a multi-target kinase stimulating indirect cell proliferation and migration. Additionally, phosphorylated VEGFR-2 induces protein kinase B (commonly known as AKT) at the beginning of PI3K/AKT/mTOR pathway, an important signalling regulator of the cell cycle and metabolism, reducing risk of apoptosis and promoting cellular transcription, proliferation and migration [58,106]. Moreover, phosphorylated VEGFR-2 activates signalling of focal adhesion kinase (FAK) observed during cellular migration, adhesion, cytoskeleton rearrangement and tumour progression [107,108]. Nevertheless, it was observed that VEGF-A can regulate endothelial cell attachment independently of VEGFR-2 through NPR-1 [109]. 

VEGF receptors, in addition to transmembrane forms, can also occur in soluble forms, known as sVEGFR-1 and sVEGFR-2 (Figure 1) [108,110]. Their formation can be explained by two mechanisms, namely, a proteolysis of extracellular binding domain [111,112] and alternative splicing of primary gene transcript [108,113], both forming freely diffusible proteins consisting of only six of seven Ig-like subunits [114]. Soluble receptors are secreted by identical cells that express regular receptors, mostly by vascular endothelial cells [110]. Due to the fact that sVEGFRs exhibit comparable binding affinity on a similar basis as regular receptors, but are deprived of effector domains of tyrosine kinases, they can demonstrate only a regulatory decoy function. Both soluble receptors compete for VEGF-A with regular receptors inhibiting angiogenic and other actions of the growth factor. Simultaneously, sVEGFR-2 can uptake VEGF-C and VEGF-D reducing their overall supply intended for lymphangiogenesis stimulation through VEGFR-3 [113]. Moreover, creation of heterodimers from soluble and regular receptors precludes cellular signalling [110]; however, it is suspected that interaction of sVEGFRs with NRP-1 can mediate VEGF-A trigger of intracellular PKC pathway signalling [115].

Interestingly, several reports have shown the reverse correlation between sVEGFR expression and cancerous angiogenesis or metastasis. Such research has indicated that sVEGFR-1 permanently suppresses tumour growth and decreases metastasis promoting overall survival rate in rodents or humans with fibrosarcoma and glioblastoma [116], advanced renal cancer [117], breast cancer [118,119], acute myeloid leukaemia [120], colorectal cancer [121] and non-small cell lung cancer [122]. Similar results were presented for sVEGFR-2 [119,123,124,125,126], demonstrating significant biomarker role of these receptors in diagnosis of numerous cancers.

### 2.3. Anti-Angiogenic Therapy Strategies for Tumour Treatment

Although various angiogenesis-stimulating factors exist, VEGF-A is considered the most potent and predominant one. This also applies to sustained angiogenesis in cancers. Currently, it is known that angiogenesis, besides its crucial role in the tumour growth, stimulates the progression of invasiveness and development of vascular network in the surrounding tumour microenvironment [127,128]. The concept of angiogenesis targeting for cancer diagnosis and treatment seems promising, therefore, a wide variety of therapeutic strategies have been directed at visualisation and interfering with tumour-stimulated angiogenesis. However, since the first FDA approval of bevacizumab (BV), humanised anti-VEGF-A mAb, for the combinational chemotherapy regimen with 5-fluorouracil of metastatic colorectal cancer [129], only a few AAT strategies have been granted similar approval. It has become a challenge to evaluate these strategies almost personally for each patient, due to considerable variability of the angiogenic process in each treated entity [42]. Although the correlation between tumour progression and VEGF-A expression is well established, it does not transfer into intended anti-angiogenic therapeutic effects. This is due to the heterogeneity of the same tumour between patients, but also between different tumours in an individual patient, that occurs and changes at different stages of the lesion development. This raises the need for appropriate methods of assessing how the patient responds to the proposed therapy. In terms of AAT, this applies to clinically significant parameters as the lesion location with regard to tumour admission of therapeutic agents and expression of endogenous growth factors in tumour microenvironment affecting the saturation of target receptors involved in angiogenesis. Despite the complexity of this issue, the use of radiopharmaceuticals is increasingly proposed for independent preliminary screening, which can provide the prediction of patient clinical response [130]. Radioligands successfully targeting VEGF/VEGFR system in vivo are potentially valuable tracers for the study of angiogenic processes [131], stratification of patients to AATs [132], as well as monitoring therapy efficacy and clinical outcomes [133,134].

Basically, the aforementioned radiopharmaceuticals are based on various approaches to VEGF/VEGFR system targeting including radiolabelled derivatives of human VEGF-A ligands, anti-VEGF or anti-VEGFR antibodies, VEGFR binding peptides, small molecular inhibitors of tyrosine kinase domain of VEGF receptors and peptidomimetic ligands targeting NRP-1 co-receptor. Additionally, depending on specific radiation features of applied radionuclide, the radiopharmaceuticals are dedicated for diagnostic, therapeutic or theranostic purposes. This multitude of radiopharmaceutical solutions allows for the design of tailor-made therapeutic tool and its evaluation on a specific cancer model. The broad selection of above listed biovectors enables choice of one that provides the desired multiple molecular targets or just specific one, exhibits eligible pharmacokinetics, predicts response of certain chemotherapeutic strategy, or shows confirmed complemental contribution to the selected chemotherapy.

AAT methods have especially found a place in clinical practice applied in monotherapy. Currently, it is well known that even these methods used alone are inefficient, they advantageously support conventional chemotherapy effects [135]. Interestingly, the AAT contributes to normalisation of the tumour vasculature resulting in enhanced metabolic rate and delivery capacity of the tumour; hence, AAT can increase efficacy of the radiotherapy or activity of immune system in the close tumour surroundings.

## 3. The Role of VEGFR and Their Ligands in Combination Targeted Radionuclide Tumour Treatment

### 3.1. Radiolabelled VEGF Ligands and Their Derivatives 

VEGF pro-angiogenic factors are one of the most often labelled compounds in angiogenesis imaging studies. Of the many known VEGF-A isoforms, mainly VEGF-A_121_ and VEGF-A_165_ are employed as a biovectors in radiotracers for imaging purposes. These compounds are generally labelled with diagnostic radionuclides dedicated for both SPECT (^99m^Tc, ^111^In, ^123^I and ^125^I) and PET (^18^F, ^61^Cu, ^64^Cu, ^68^Ga, ^89^Zr). The literature shows that there are also descriptions of therapeutic radionuclide (^177^Lu and ^188^Re) in labelled VEGF derivatives [136,137,138]. VEGF-A-based radiocompounds are used extensively for selective imaging of VEGFR overexpressions, but also, inter alia, for therapy monitoring [139], new tracers search [140], estimation of tumour vascularity [141], correlation between receptor density and disease progression [142] or as a prognostic marker for treatment progress evaluation [143]. 

The first reports on radiolabelled derivatives of VEGF-A generally focused on the use of radioiodinated VEGF-A isoforms, [^125^I]I-VEGF-A_121/165_, for pharmacokinetic and pharmacodynamic studies on various cell lines, animal models, as well as autoradiography studies on human cancers [91,144,145,146,147,148,149]. 

Unmodified VEGF-A_121_ and its derivates were most often labelled with copper-64 ([^64^Cu]Cu-DOTA-VEGF-A_121_) [132,133,150,151,152,153], gallium-68 ([^68^Ga]Ga-NOTA-VEGF-A_121_ and [^68^Ga]Ga-NODAGA-VEGF_121_) [154,155], iodine-123 and iodine -125 (([^123/125^I]I-VEGF-A_121/165_) [141,156] and in one case with indium-111 ([^111^In]In-DTPA-VEGF-A_121_) [157]. 

[^64^Cu]Cu-DOTA-VEGF-A_121_ was applied for the first time by Cai et al., where the inverse correlation between radiocompound accumulation and size of human glioblastoma xenografts in mice was demonstrated [133]. Due to high [^64^Cu]Cu-DOTA-VEGF-A_121_ uptake in kidney, Wang et al. proposed another radiotracer, [^64^Cu]Cu-DOTA-VEGF_DEE_ [132]. Both radiocompounds exhibited similar uptake in tumour and major organs with exception of the kidneys, where [^64^Cu]Cu-DOTA-VEGF_DEE_ uptake was almost 2-fold lower. 

More detailed studies on the previously found by Cai et al. correlation [133] was carried out with the use of the same [^64^Cu]Cu-DOTA-VEGF-A_121_ radiocompound by Chen et al. [153]. In this study, the highest tracer uptake was observed in medium tumours (100-250 mm^3^) and correlated with the highest receptor expression, which was determined using specific anti-VEGFR-2 antibody staining on tested tumours [153]. 

The next three works focused on the use of [^64^Cu]Cu-DOTA-VEGF-A_121_ and/or [^64^Cu]Cu-DOTA-VEGF_mutant_ to study the level of VEGFR expression in post-stroke angiogenesis in rats [150], peripheral arterial disease in murine hindlimb ischemia [151] and myocardial infarctions in rats [152]. All reports showed specific uptake in ischemic tissues; however, superior imaging quality was provided by [^64^Cu]Cu-DOTA-VEGF-A_121_ radioligand [150,151].

Radiocompounds with gallium-68 were demonstrated in two reports [154,155]. Kang et al. used [^68^Ga]Ga-NOTA-VEGF-A_121_ for imaging VEGFR expression in U87MG cell line xenograft murine models [154]. This radiocompound showed relatively low tumour affinity and high uptake in the liver and spleen, thereby poor imaging ability in glioblastoma neoangiogenesis. A similar study showed that a more hydrophilic radiocompound [^68^Ga]Ga-NODAGA-VEGF-A_121_ also gave poor results [155]. 

VEGF-A_121_ was also labelled with ^111^In ([^111^In]In-DTPA-VEGF-A_121_) to detect VEGFR expression in a rabbit model in hindlimb ischemia [157]. Despite the pronounced radiotracer uptake in ischemic muscle, the very high accumulation in other organs limit the application of this radioagent. 

Iodinated radiocompounds were studied by Li et al., where [^123^I]I-VEGF-A_121_ and [^123^I]I-VEGF-A_165_ radiocompounds successfully demonstrated the overexpression of VEGF specific binding sites on various types of tumour cells in comparison to normal cell lines [156]. [^123^I]I-VEGF-A_165_ binds to more types of tumour cell due to affinity to both VEGFR-1 and VEGFR-2, according to data presented in cited reports [100,158]. Yoshimoto et al. compared the action of the same ligands but labelled with ^125^I, [^125^I]I-VEGF-A_121_ and [^125^I]I-VEGF-A_165_, on LS180 tumour xenograft murine model [141]. [^125^I]I-VEGF-A_121_ proved to be a more promising tumour imaging agent due to its enhanced specific tumour accumulation related to high binding affinity towards VEGFR-2.

Studies on modified VEGF-A_121_ isoform focused on the use of a recombinant VEGF protein with Cys-tag motif labelled with various radionuclides (^18^F, ^64^Cu, ^68^Ga, ^89^Zr, ^99m^Tc and ^177^Lu) [134,136,137,139,140,142,159,160,161,162,163,164,165,166,167]. This cysteine-containing fusion tag motif is commonly used for site-specific protein conjugation without affecting vector functionality and was evaluated based on synthesis and study of [^99m^Tc]Tc-scVEGF-PEG-DOTA, [^99m^Tc]Tc-HYNIC-scVEGF and [^64^Cu]Cu-DOTA-PEG-scVEGF radiotracers [160].

The Cys-tag motif was used for the first time in [^99m^Tc]Tc-HYNIC-C-tagged-VEGF synthesis and the obtained radiocompound was compared with [^99m^Tc]Tc-HYNIC-biotin-inactivated-VEGF for tumour vasculature imaging of 4T1 murine tumours [159]. The obtained results showed that [^99m^Tc]Tc-HYNIC-C-tagged-VEGF exhibited higher tumour uptake compared to that of inactivated-VEGF radioagent. Similar results were presented by Backer et al. when examining [^99m^Tc]Tc-HYNIC-scVEGF and [^64^Cu]Cu-DOTA-PEG-scVEGF [134]. Based on the results of biodistribution study on mice model, the authors observed the superiority of [^64^Cu]Cu-DOTA-PEG-scVEGF radiocompound due to more favourable pharmacokinetics compared to that of [^99m^Tc]Tc-HYNIC-scVEGF; however, both tracers showed only detectable and generally heterogenous tumour accumulation.

The research group of Levashova described the usefulness of ^99m^Tc directly-labelled single-chain VEGF (scVEGF) complexed with tricine—the synthesis of which, by definition, should be easier and faster [139,161,162]. The obtained [^99m^Tc]Tc-scVEGF was used: (1) to compare its tumour and non-tumour uptake with that of chelator-containing radioconjugates (e.g., [^64^Cu]Cu-DOTA-PEG-scVEGF and [^99m^Tc]Tc-HYNIC-scVEGF [134]) [161]; (2) to compare its effectivities in imaging of thigh abscesses in mouse model with that of [^99m^Tc]Tc-inactivated-VEGF [162]; (3) imaging of VEGFR expression changes in breast cancer xenograft mice under sunitinib treatment [139]. These studies have shown the practical performance of direct labelling, as well as the general usefulness of radiocompounds obtained by this method for imaging of thigh abscesses [162] or changes in VEGFR-2 prevalence in tumour [139]. Additionally, reports have shown that studied [^99m^Tc]Tc-scVEGF radiocompound, similar to radiocompounds with chelators in their structure, exhibits high kidney and liver uptakes with relatively low tumour uptake [161]. Blankenberg et al. presented the evaluation of efficiency of RTK inhibitor pazopanib treatment on HT29 xenografts in mice, which was assessed through VEGFR-2 imaging using [^99m^Tc]Tc-scVEGF radioagent [163].

Comprehensive research on scVEGF labelling with ^68^Ga invastigated the use of various reactions parameters: chelator types, lengths of PEG linker and syntheses conditions [164,165]. [^68^Ga]Ga-HBED-CC-PEG-scVEGF and [^68^Ga]Ga-NOTA-PEG-scVEGF were studied by Eder et al. [164] and [^68^Ga]Ga-NOTA-PEG-scVEGF and [^68^Ga]Ga-DOTA-PEG-scVEGF by Blom et al. [165] as VEGFR-2 imaging radiotracers in various human cell line xenografts in mice. Firstly, Eder et al. showed that stability, binding assay, biodistribution and PET imaging were similar for both radioagents; however, [^68^Ga]Ga-HBED-CC-PEG-scVEGF radiotracer possessed a more effective and faster labelling end point, which was advantageous for compound labelling with short-lived radionuclides. Blom et al. demonstrated the advantage of microwave heating synthesis over conventional synthesis, as well as there were no impact on different linker lengths [165].

The possibility of selective imaging of VEGFR-1 and VEGFR-2 receptors on the tumour surface and in various diseases, discussed in the following articles, may provide detailed information on overexpression of receptors and thus allow more accurate imaging of specific therapy solutions. Tekabe et al. used technetium-99m labelled scVEGF-PEG-DOTA [142] and its mutant versions, scVR1-PEG-DOTA and scVR2-PEG-DOTA, which have selective affinity to VEGFR-1 and VEGFR-2, respectively [167]. In both studies, the obtained results confirmed the expected usefulness of radiotracers in plaque progression monitoring of atherosclerosis in diabetic mice. Another efficient imaging of VEGFR-1 or VEGFR-2 was confirmed on breast cancer mice model using [^89^Zr]Zr-DFO-PEG-scVR1 and [^89^Zr]Zr-DFO-PEG-scVR2 radiocompounds [166].

[^18^F]FBEM-scVEGF radioagent was dedicated for VEGFRs imaging in mouse models with xenografts of various tumour cell lines (MDA-MB-435, U87MG and 4T1) [140]. The results of performed tests showed VEGFR specific tumour uptake, as well as significant uptake in kidneys, lung and intestine. Therefore, the authors qualified this radiotracer for VEGFR-2 overexpressing tumour imaging.

scVEGF-based radiotracer containing therapeutic radionuclide lutetium-177, [^177^Lu]Lu-DOTA-PEG-scVEGF, was studied as a radioagent for targeted systemic radiotherapy by Blakenberg et al. [136] and in combined chemotherapy by Rusckowski et al. [137]. Blakenberg et al. compared the tumour growth inhibition effectiveness of three radiocompounds with different lengths of PEG linkers (2.0, 3.4 and 5.0 kDa). Based on the experimental results, the authors concluded that tumour growth inhibition was dose dependent and that there were significantly different therapy responses between individual MDA231luc tumour bearing mice. Moreover, the most promising radiocompound (containing the linker 3.4 kDa PEG) was established, as well as its well-tolerated single dose necessary to obtain the tumour growth inhibition effect. Further studies on the effectiveness of [^177^Lu]Lu-DOTA-PEG-scVEGF alone or with doxorubicin were performed on mice with metastatic and orthotopic triple-negative breast cancer [137]. The results showed inhibition of tumour metastasis and prolongation of the survival time in the case of mice treated with [^177^Lu]Lu-DOTA-PEG-scVEGF compared to the control group of mice treated with scVEGF-PEG-DOTA. Orthotopic tumours displayed superiority of combination therapy (compared to [^177^Lu]Lu-DOTA-PEG-scVEGF or doxorubicin used separately) with a noticeably longer tumour doubling time.

The application of [^61^Cu]Cu-NOTA-K3-VEGF-A_121_ radiocompound (based on VEGF-A_121_ modified with three lysine residues, K3-VEGF-A_121_) for PET/CT imaging of VEGFR expression on 4T1 tumour-bearing mice was described by Zhang et al. [168]. The experimental results clearly showed noticeable radiotracer uptake in tumour, but also significant uptake in the liver. The authors indicated that [^61^Cu]Cu-NOTA-K3-VEGF-A_121_ would be a promising radiocompound to imaging VEGFR expressions.

Numerous reports have utilised modified VEGF-A_121_ protein in multimodal VEGFR imaging studies [169,170,171]. Blankenberg et al. described the synthesis of [^99m^Tc]Tc-HuS/Hu-VEGF and [^99m^Tc]Tc-HuS/Hu-P4G7 (the latter based on anti-VEGFR-2 single-chain antibody), which were both evaluated during VEGFR-2 expression imaging of subcutaneous and pulmonary 4T1 luc/gfp adenocarcinoma tumours [169]. VEGFR-2 imaging and biodistribution studies on tumour-bearig mice showed higher uptake of [^99m^Tc]Tc-HuS/Hu-VEGF in comparison to that of [^99m^Tc]Tc-HuS/Hu-P4G7. Hence, the authors recommended [^99m^Tc]Tc-HuS/Hu-VEGF radioagent for imaging of small subcutaneous and internal VEGFR-2 overexpression tumours.

The examples of VEGF-based radiotracer application for dual-modality imaging, PET and near-infrared fluorescent (PET/NIRF) were described by Chen et al. [170] and Kang et al. [171]. The first approach consisted of DOTA and VEGF-DOTA conjugate that were coupled with amine-functionalised quantum dots (QD) and labelled with copper-64 radionuclide. In vitro and in vivo studies of [^64^Cu]Cu-DOTA-QD and [^64^Cu]Cu-DOTA-QD-VEGF displayed VEGFR-specific binding and uptake in U87MG tumour only for VEGF-based radioagent, as well as correlation between results obtained in PET and NIRF methods. The second approach described similar studies (VEGFR expression imaging in U87MG tumour-bearing mice) using [^64^Cu]Cu-DOTA-(AF)-SAv/biotin-PEG-VEGF-A_121_ radiotracer (detailed description of the synthesis and labelling procedure was presented by Kang et al. [171]). Radiochemical yield and specific activity of synthesised radiotracer were generally low (31.40 ± 3.30% and 1.96 ± 0.67 GBq/mg, respectively), whereas its tumour uptake and stability in 50% fetial bovine serum at 37 °C for 24 h (>93%) were satisfactorily high. Unfortunately, high radioactivity accumulation in ex vivo and microPET studies was observed also in the liver and kidneys. The remarkably high uptake in the same U87MG tumours was observed in [^64^Cu]Cu-DOTA-VEGF-A_121_ radiotracer [133], however, it should be noted that these two studies evaluated different tumour sizes [133,169].

Another modified VEGF-A_121_ derivative labelled with copper-64 radionuclide was described by Hsu et al. [172]. The research studied tumour uptake and VEGFR imaging in mice with implemented U87MG-fLuc cells, where [^64^Cu]Cu-DOTA-VEGF-A_121_/rGel radioagent was used as an efficient marker to evaluate the results of VEGF-A_121_/rGel anti-angiogenic and anti-tumour treatment. Detailed information regarding VEGF-A_121_/rGel toxin and its activities are available in references [173,174,175,176].

Clinical applications of VEGF-A_165_-based radiocompound, [^123^I]I-VEGF-A_165_ for scintigraphic tumour localisation was evaluated in patients with gastrointestinal tumours [177], pancreatic carcinoma [178], highly malignant osteosarcoma [179] and various grades primary brain tumours [143]. The results showed fast and different uptake in various lesions; however, contrary to CT and MRI, SPECT imaging detection of existing tumours and metastases was the least sensitive [177]. [^123^I]I-VEGF-A_165_ radiocompound used for preliminary scintigraphy of highly malignant VEGFR-positive osteosarcoma gave very promising results; however, due to an unrepresentative number of two patients, they may require further verification [179]. The differences in [^123^I]I-VEGF-A_165_ uptake and survival time in patients with various grades and sizes of primary brain tumours were studied by Rainer et al. [143]. The obtained scintigraphy results showed correlation between the tumour grade and radiotracer uptake, which allowed distinguishability between VEGFR-positive (stage IV) and VEGFR-negative (stage II and III) tumours. Conrelissen et al. evaluated [^123^I]I-VEGF-A_165_ and [^125^I]I-VEGF-A_165_ on melanoma tumour-bearing mice for future assessment of efficiency of farnesyl transferase inhibitors therapy [180]. The results revealed low, but noticeable radiotraces tumour uptake, but, also significant uptake in kidneys, intestine and stomach in the case of both control and A2058 tumour-bearing athymic mice [180]. Nevertheless, it was concluded that [^123^I]I-VEGF-A_165_ radiotracer could be a potential tool for imaging of VEGFR overexpression.

[^99m^Tc]Tc-HYNIC-VEGF-A_165_ radiocompound with unmodified VEGF-A_165_ was evaluated for VEGFR expressions in various xenografts tumours in mice [181]. The authors admitted that studies with VEGF-based radiocompounds gave false negative results in the case of large tumours due to receptor saturation by in situ secreted VEGF glycoproteins. Similar observations were presented earlier by Cai et al., where larger tumour showed lower uptake of radiolabelled [^64^Cu]Cu-DOTA-VEGF-A_121_ than that of smaller size [133].

Three recombinant VEGF-A_165_ derivatives: human transferrin-VEGF-A_165_ (hnTf-VEGF-A_165_), VEGF-2K and VEGF-2K-NLS labelled with indium-111 or technetium-99m were described by Chan et al. [182,183]. In the first study the distribution of [^111^In]In-hnTf-VEGF-A_165_ in athymic mice bearing U87MG human glioblastoma xenografts, performed in excess of apotransferrin and VEGF-A_165_, showed specific binding of radioagent towards VEGFRs [182]. The second study focused on the comparison of the cytotoxicity of [^111^In]In/[^99m^Tc]Tc-DTPA-VEGF-2K and [^111^In]In/[^99m^Tc]Tc-DTPA-VEGF-2K-NLS radiocompounds on PAE cells overexpressing VEGFR-1 [183]. Based on the study results, the authors concluded that [^111^In]In-DTPA-VEGF-2K-NLS radiotracer exhibited the greatest cytotoxic properties.

*VEGFA* gene exon 6-encoded peptide, QKRKRKKSRYKS and its derivatives, QKRKRKKSRKKH and RKRKRKKSRYIVLS, exhibit an anti-angiogenic effect due to the binding to VEGFRs without activation of VEGFR TKs, and were able to compete with endogenous VEGF for the receptor binding site [138,184]. The first peptide labelled with rhenium-188, [^188^Re]Re-MAG_3_-QKRKRKKSRYKS, was used to compare the results of ex vivo distribution and SPECT tumour imaging regarding VEGFR-2 [184]. Experimental results confirmed higher accumulation in truncated receptor tumours than for full length ones. Zhang et al. described two novel radiotracers, QKRKRKKSRKKH and RKRKRKKSRYIVLS, derived from VEGF-A_125-136_, which exhibited significantly higher affinity to VEGFR-1 than QKRKRKKSRYKS. These peptides were labelled with technetium-99m or rhenium-188 using HYNIC or ethylene dicysteine (EC) chelators in order to study potential radioagents for tumour diagnosis and therapy [138]. The obtained radiotracers [^99m^Tc]Tc-HYNIC-QKRKRKKSRKKH, [^99m^Tc]Tc-HYNIC-RKRKRKKSRYIVLS, [^188^Re]Re-EC-QKRKRKKSRKKH and [^188^Re]Re-EC-RKRKRKKSRYIVLS showed high stability in saline and human serum and high specific accumulation in tumour sites on mice with A549 tumour xenografts. In the case of ^188^Re-labelled peptides, the authors also recorded a noticeable therapeutic effect. According to the authors conclusion, all studied radiopreparations were considered promising candidates for tumour radionuclide imaging and therapy.

The wide range of works concerning VEGF protein labelled with diagnostic or therapeutic radionuclides presented in this paragraph indicates the significant evolution of this field of research. The VEGF protein was used both in unmodified and modified forms, mainly as single-chain recombinant VEGF with a cysteine-containing fusion tag (C-tag), which allows site-specific conjugation without affecting protein functionality. Reports have confirmed that such modification maintains functional characteristics of proteins [134,159]. Moreover, many articles found that C-tagged-radiopeptides shows greater accumulation in VEGFR-2 expressed areas compared to that of their inactivated forms [134,142,159,162,164]. Unfortunately, significant limitations in the use of VEGF-based radiocompounds reside in their high uptake mainly in the kidneys and liver, which hinders proper visualisation of the tumours. In the cited articles, none of the studies refered to the reduction of renal uptake, e.g., using infusion of amino acids [185] or Gelofusine preparation [186,187]. The poor imaging results are also influenced by uneven VEGF expression on the surface of tumours, related to tumour size or grade, and the phenomenon of receptor saturation by endogenous VEGF. An interesting method for VEGFR-1 and VEGFR-2 imaging may be the application of radiotracers [^89^Zr]Zr-DFO-PEG-scVR1/scVR2 [166] or [^99m^Tc]Tc-scVR1/scVR2-PEG-DOTA [167], capable of selective and separate imaging of each receptor. It is also worth mentioning that short peptides QKRKRKKSRYKS, QKRKRKKSRKKH and RKRKRKKSRYIVLS can be successfully utilised for the design and synthesis of VEGF-based radiotracers [138,184]. In summary, unfortunately, it seems that most of the above disscussed VEGF-based radiopreparations do not fulfil the requirements for VEGFRs imaging agents. Concise information concerning radiolabelled VEGF ligands and their derivatives is presented in Table 1.

### 3.2. Radiolabelled Anti-VEGF and Anti-VEGFR Antibodies

Anti-VEGF antibodies belong to a group of angiogenesis inhibitors acting in an indirect manner. They operate by interfering with the pro-angiogenic communication between the tumour cells and endothelial cell by blocking VEGF ligands binding to their receptor. As a result, occurrence of VEGF-induced proliferation, permeability, survival and growth of endothelial cells is restricted. Radiolabelled anti-VEGF antibodies can be used, both alone or in combination with chemotherapy in targeted radionuclide tumour therapy, for tumour uptake imaging and can play the role of predictive markers. The following works detail the landscape of current literature regarding various anti-VEGF antibodies labelled with radionuclides, which are arranged in chronological manner corresponding to publishing of performed studies.

The first reports examine the use of anti-VEGF or anti-VEGFR antibodies labelled with various radionuclides in nuclear medicine concern HuMV833, VG76e, Avas12a1 and DC101 mAbs.

HuMV833 anti-VEGF antibody (mean biological half-life 8–9 days), a humanised IgG4ĸ mAb that binds VEGF-A_121_ and VEGF-A_165_ and has anti-tumour activity against a wide spectrum of human tumour xenografts, was the first evaluated anti-VEGF mAb in preclinical studies [188]. Binding studies of [^124^I]I-HuMV833 radioconjugate, showed high affinity for VEGF and confirmed that VEGF-binding potential was retained after iodination [26,46,188,189,190,191,192]. The authors also highlighted the issue of significant differences in radioconjugate tumour uptake, radioconjugate clearance and biologic response during and after therapy ocurrig even within the same tumour among patients and the different tumours within the same patient.

Collingridge et al., Bouziotis et al. and Fani et al. described radiopreparations based on anti-VEGF antibody VG76e [193,194,195]. VG76e, a mouse IgG_1_ mAb capable of recognising 121, 165 and 189 isoforms of human VEGF, was recommended for detection of VEGFs being the prognostic indicators of cancer occurrence [46,131,189,190,191,193,194,195]. Collingridge et al. using different iodination methods synthesised and tested four VG76e-based radioconjugates: [^125^I]I-VG76e, [^125^I]I-SIB-VG76e, [^125^I]I-SHPP-VG76e for SPECT imaging and [^124^I]I-SHPP-VG76e for PET imaging of human fibrosarcoma cell line HT1080 xenografts in mice. Radioconjugate [^124^I]I-SHPP-VG76e displayed so satisfactory results, that the authors proposed its application for classification of patients for anti-angiogenic therapy, investigation of angiogenesis pathways in vivo and determination of resistance mechanisms of AAT agents.

Bouziotis et al. studied the same anti-VEGF antibody labelled with technetium-99m, samarium-153 and lutetium-177 [194], whereas Fani et al. applied lutetium-177, using DOTA and DTPA chelating agents [195]. Biodistribution studies of [^99m^Tc]Tc-VG76e, [^153^Sm]Sm-DTPA-VG76e and [^177^Lu]Lu-DTPA-VG76e radiotracers were performed on mice with human breast adenocarcinoma cell line MCF7 xenografts. Research showed that VEGFs could be effectively visualised using VG76e antibody labelled with diagnostic radionuclides [194]. Both therapeutic radioconjugates [^177^Lu]Lu-DOTA-VG76e and [^177^Lu]Lu-DTPA-VG76e showed similar and satisfactory physicochemical and biological properties, which indicated that they could be successfully employed in tumour radioimmunotherapy [195].

Willmann et al. described the application of [^125^I]MBs-I-Bt-Avas12a1 [196] and [^18^F]MBs-SFB-Avas12a1 [197] radioconjugates, containing rat anti-VEGFR-2 mAb, Avas12a1, for tumour angiogenesis imaging using ultrasonographic (US) or dual-model US/PET approaches. [^125^I]MBs-I-Bt-Avas12a1 was used only as a radiotracer in the coupling reaction of streptavidin-containing microbubbles (MBs) and iodine-125 labelled biotinylated anti-VEGFR-2 monoclonal antibody Avas12a1 (Bt-Avas12a1) for the quantitative assessment of reaction products [196]. The second radiocompound, [^18^F]MBs-SFB-Avas12a1, formed in the reaction of MBs and the radiofluorine agent [^18^F]N-succinimidyl-4-fluorobenzoate (SFB) coupled with Avas12a1 antibody, was used for dynamic micro-PET imaging on nude mice bearing angiosarcomas. It was concluded that fluorine radiotracer allowed non-invasive assessment of the whole-body biodistribution with targeted VEGFR-2 radiotracer in most tissues of living mice and could be used for studying the biology of angiogenesis in living subjects [197].

Lee et al. presented another radioagent based on rat anti-VEGFR-2 mAb DC101, dedicated for ischemia monitoring, containing chitosan polymer and diagnostic radionuclide Tc-99m [198]. Examination of [^99m^Tc]Tc-HYNIC-chtiosan-Cy5.5-DC101 was performed on human umbilical vein endothelial cells and mice with surgically induced ischemia. The study revealed good correlation between radiotracer uptake in ischemic tissues and changes in VEGFR-2 expression, which proved that the radiocompound could be successfully used for imaging of ischemic areas [198].

The most widely used anti-VEGF antibody in cancer diagnosis and therapy is bevacizumab (BV, mean biological half-life is 17–21 days). This humanised mAb, sold under the brand name Avastin, is a medication used in combinational immunotherapy to treat numerous types of cancers (e.g. colon cancer [24], lung cancer [24], breast cancer [24], renal-cell carcinoma [24,199,200], glioblastoma multiforme [201,202,203,204]) as well as age-related macular degeneration [6,23]. It binds to all isoforms of VEGF and thus prevents interactions with VEGFR-1 and VEGFR-2 [41].

Various research groups tested BV labelled with numerous diagnostic (^64^Cu, ^86^Y, ^89^Zr, ^99m^Tc, ^111^In, ^124^I, ^125^I) and therapeutic (^90^Y, ^131^I, ^177^Lu) radionuclides and BV-based radioconjugates, in general, for non-invasive measurement of VEGF expression in tumours both prior to the initiation of anticancer therapy and during therapy conducted with different medications [191,199,205,206,207,208,209,210,211,212,213,214,215,216,217,218,219,220,221,222,223,224,225,226,227,228,229,230,231,232,233,234,235,236,237,238,239].

Nagengast et al. examined labelling reactions of BV with In-111 (using DTPA chelator) and Zr-89 (using N-succinyl-desferrioxamine (N-suc-Df) chelator), as well as stability of [^111^In]In-DTPA-BV and [^89^Zr]Zr-N-suc-Df-BV radioconjugates [205]. Comparison of the binding studies results of BV-based radioconjugates and human [^89^Zr]Zr-N-suc-Df-IgG as a nonspecific control in SKOV-3 ovarian tumour-bearing mouse model showed that these new BV-based radioconjugates could be used as tracers for non-invasive imaging of VEGF in tumour microenvironment during anti-angiogenic therapy [45,46,47,191,192,205,212,240,241]. However, the study revealed that despite the success of AAT, some patients treated with BV did not benefit from this targeted therapy, possibly due to BV failure in reaching the target.

Sheer et al. and Nagengast et al. focused on correlation between [^111^In]In-DTPA-BV radioconjugate tumour uptake and the level of cancer VEGF-A expression [206,207]. Research of patients with liver metastases of the colon did not show such correlation [26,45,46,192,206]; however, it was noticed during VEGF imaging of the melanoma lesions treated with BV [46,207,241].

Scintigraphical visualisation of VEGF-A expression in tumours in mice with xenografts of human colon carcinoma LS174T cell line using [^111^In]In-DTPA-BV and [^125^I]I-BV radioconjugates were presented by Stollman et al. [208]. Experimental results showed that uptake of [^111^In]In-DTPA-BV was significantly higher than that of [^125^I]I-BV. According to author’s evaluation, both radiocompounds were regarded as prognostic factor. Imaging of VEGF-A expression provided information on prognosis and response to chemotherapy and allowed patient classification for anti-VEGF AAT.

Stollman et al. also investigated [^111^In]In-DTPA-BV radiocompound uptake in Mel57 human melanoma cell line expressing different VEGF-A isoforms (121, 165 and 189). High specific tumour uptake of the labelled antibody was detected only in the case of VEGF-A_165_ and VEGF-A_189_ isoforms [209]. Furthermore, it was suggested that scintigraphy imaging of these VEGF isoforms could be useful for predicting responses to AAT.

Scintigraphy imaging of renal cell cancer and evaluation of neoadjuvant treatment with VEGFR inhibitor sorafenib were studied by Desar et al. using [^111^In]In-DTPA-BV [199]. Based on the experimental results, authors concluded that cancer progression can be effectively visualised using this radioconjugate and that neoadjuvant treatment significantly decreases [^111^In]In-DTPA-BV cancer accumulation. However, the reduced uptake of the radioconjugate was not the consequence of reduced VEGF-A expression (VEGF-A expression was measured independently by ELISA method), indicating, similarly as in the case of Sheer’s study [206], that there was no correlation between [^111^In]In-DTPA-BV tumour uptake and VEGF expression. Nevertheless, [^111^In]In-DTPA-BV scintigraphy could be considered an attractive biomarker of therapy response; however, this approach requires further research.

BV-based radioconjugates containing diagnostic radionuclides In-111 and Y-86 and therapeutic radionuclide Lu-177 were synthesised and tested by Hosseinimehr et al., Nayak et al., and Kameswaran et al. [210,211,213]. Binding specificity study of [^111^In]In-CHX-A″-DTPA-BV showed high specific binding towards human ovary SKOV-3 and colon LS174T tumour cell lines, as well as non-specific binding towards prostate cancer DU 145 cell line [210]. This radioconjugate was dedicated for scintigraphic imaging of VEGF expression before patient selection for anti-VEGF therapy. Biodistribution and PET imaging studies of [^86^Y]Y-CHX-A″-DTPA-BV were performed on mice bearing VEGF-A negative human mesothelioma MSTO-211H and VEGF-A secreting LS-174T and SKOV-3 xenografts [46,47,211]. The preclinical study of the radioconjugate demonstrated its potential for non-invasive assessment of VEGF-A tumour angiogenesis status and possibile application as a marker in radioimmunotherapy conducted using therapeutic radioconjugate [^90^Y]Y-CHX-A″-DTPA-BV. Binding studies of [^177^Lu]Lu-CHX-A″-DTPA-BV were performed using VEGF expressing U937 tumour cell line [213]. The studied therapeutic radiopharmaceutical showed high in vitro stability, strong cell binding, as well as high and specific uptake by VEGF overexpressing melanoma cells.

Patel et al. described the specificity and pharmacokinetics of [^111^In]In-DTPA-BV binding to VEGF and its use for assessment of response to rapamycin inhibition of mTOR kinase used in the treatment of renal and breast cancer [214]. Clinical trials in patients with renal cell carcinoma and metastatic colorectal cancer showed no correlation, similar to the previously discussed reports [199,206], between uptake of radiotracer and intratumoural VEGF expression, determined by ELISA assay, in situ hybridisation or immunohistochemically.

Yudistiro et al. investigated the potential application of novel BV-based radioconjugates (biotinylated-BV, Bt-BV) [215]. In general, the disadvantage of BV radioimmunotherapy is its low clearance value, which causes high risk of haematotoxicity. The authors synthesised new radioconjugates [^111^In]In-DTPA-Bt-BV and [^90^Y]Y-DTPA-Bt-BV and tested their biological properties in mice triple negative breast cancer xenograft model. Application of avidin chase strategy for [^90^Y]Y-DTPA-Bt-BV therapy increased the maximum tolerated dose of the therapeutic radioconjugate and consequently improved the therapeutic outcome.

Utilisation of BV labelled with zirconium-89 for VEGF level imaging was the subject of many reported experimental and clinical studies [26,46,47,216,240]. Based on the knowledge that in animal models targeted heat shock protein 90 inhibition therapy leads to reduction of VEGF secretion and mean vascular density in tumour cell lines, Nagengast et al. examined the possibility of PET imaging using [^89^Zr]Zr-N-suc-Df-BV for in vivo non-invasive visualisation of early changes in VEGF levels during treatment with new synthetic inhibitor NVPAUY922 [216].

Application of [^89^Zr]Zr-N-suc-Df-BV radioconjugate as an early biomarker of AAT with everolimus is described in several reports [47,217,218,219,241]. On the basis of experimental results, authors showed that everolimus, an inhibitor of mTOR pathway often activated in ovarian tumours [217], advanced neuroendocrine tumours [218] and metastatic renal cell carcinoma (RCC) [219], can reduce the production of VEGF-A stimulated by cancer cells, which in turn allows monitoring of effects on everolimus treatment using [^89^Zr]Zr-N-suc-Df-BV radiotracer by PET imaging.

Other reports by Gaykema et al. and Bahce et al. employed [^89^Zr]Zr-N-suc-Df-BV radioconjugate to image VEGF-A level in primary breast cancer patients and non-small cell lung cancer (NSCLC) patients, respectively [46,47,220,221,241]. In both cases [^89^Zr]Zr-N-suc-Df-BV uptake correlated with the level of VEGF-A in studied tumours. Moreover, VEGF-A imaging with [^89^Zr]Zr-N-suc-Df-BV radioconjugate applied in phase III study showed survival benefits for NSCLC patients treated with combination carboplatin-paclitaxel-bevacizumab regimen compared to patients treated with only these chemotherapeutics.

Potential application of [^89^Zr]Zr-N-suc-Df-BV radioconjugate for monitoring VEGF-A level changes before and during anti-angiogenic treatment with BV and IFNα and sunitinib of metastatic RCC patients and patients with VHL disease are presented in several papers published by Oosting’s research group [222,223,224]. The pilot study revealed differences in [^89^Zr]Zr-N-suc-Df-BV tumour uptake after BV/IFNα and sunitinib therapy (BV/IFNα strongly decreases [^89^Zr]Zr-N-suc-Df-BV tumour uptake, whereas sunitinib results in modest reduction), which indicated that they induce different angiogenic responses [47,222,223]. Hence, [^89^Zr]Zr-N-suc-Df-BV radioconjugate may be an effective tool, for anti-VEGF therapy stratification of patients with VHL-associated lesions [224,241].

Various BV labelling reactions with technetium-99m and application of prepared [^99m^Tc]Tc-radiopharmaceuticals for targeted nuclear medicine were reported in several publications [225,226,227]. These potential diagnostic radiopharmaceuticals were dedicated for scintigraphic imaging of VEGF levels in tumour and its environment. In general, these radiocompounds exhibited satisfactory stability in saline and serum solutions; however, significant signals related to [^99m^Tc]Tc-HYNIC-BV degradation occurred in the presence of great excess of competing ligand cysteine [225,227] and in the case of [^99m^Tc]Tc(CO)_3_-BV—in the presence of histidine [226]. Biodistribution and scintigraphy imaging using [^99m^Tc]Tc-HYNIC-BV and [^99m^Tc]Tc(CO)_3_-BV tracers were performed on mice bearing breast adenocarcinoma tumour xenografts [225] or mouse model with subcutaneous melanoma xenograft [226,227]. The observed body retention of [^99m^Tc]Tc-HYNIC-BV and [^99m^Tc]Tc(CO)_3_-BV radiocompounds indicated a mixed hepatobiliary/renal clearance. The authors suggested that all studied radiotracers could be observed in preclinical studies as a clinical tool for solid tumours screening and as markers providing response to BV chemotherapy before and after therapy. Kameswaran et al. examined BV radiolabelling reaction utilising p-SCN-Bn-DTPA chelator [228]. [^99m^Tc]Tc-DTPA-BV radiotracer was stable in histidine solution and biodistribution studies performed on murine melanoma model showed good radiotracer specificity of VEGF binding, indicating its potential as a radioimmunoscintigraphy agent for various cancers.

Cohen et al. described the procedure for dual labelling of antibodies BV and cetuximab with NIRF dye IRDye800CW and zirconium-89 for optical and PET imaging, respectively [229,230]. Biodistribution studies of [^89^Zr]Zr-N-suc-Df-BV/cetuximab-800CW and [^89^Zr]Zr-N-suc-Df-BV/cetuximab radioconjugates were performed using mice bearing human squamous carcinoma cell line A431 or human epithelial cell line FaDu and control mice. Both radioconjugates were dedicated for early photo- and radio-detection of small, established tumours that could not be identified by other radiological and nuclear techniques.

Potential application of [^89^Zr]Zr-N-suc-Df-BV radioconjugate for diagnosis in nuclear medicine was described by Jansen et al. and Veldhuijzen van Zanten et al. [231,232,233]. The possibility of [^89^Zr]Zr-N-suc-Df-BV radioconjugate application for adult and adolescent recurrent high-grade gliomas imaging was tested, including end stage diffuse intrinsic pontine glioma (DIPG) tumours. DIPG tumours, overexpressing pro-angiogenic factors, including VEGF, were resistant to various types of systemic therapies, including targeted therapies. However, the obtained results were inconclusive, no significant uptake of [^89^Zr]Zr-N-suc-Df-BV in the intracranial tumour models occurs at any stage of the disease, but the mice model displayed high and moderate uptake of the radiotracer in E98 and HSJD-DIPG-007 xenografts, respectively [231]. Moreover, authors emphasised that anti-VEGF therapy could induce more diffused and distant spread of tumour cell, hence, BV treatment was only justified if targeting of VEGF by BV as previously verified by immuno-PET scan [231]. Immuno-PET in children and adults with DIPG showed large variability in the intratumour [^89^Zr]Zr-N-suc-Df-BV uptake, suggesting large differences in local expression of VEGF within the tumour [232]. PET imaging study combined with autopsy study data confirmed high inter- and intrapatient DIPG heterogeneity, which could explain lack of benefit from BV anti-VEGF treatment in some DIPG patients [233]. Based on the conducted studies, the authors suggested that the combination of MRI and PET imaging may help in selecting potential DIPG patients for BV treatment.

Current literature has relatively little information regarding BV labelling with other radionuclides and application of such radioconjugates in nuclear medicine. Stollman et al. described the potential application of BV labelled with diagnostic radionuclide I-125, for scintigraphic visualisation of VEGF-A expressing tumours [208]. The recorded lower concentration of [^125^I]I-BV radiotracer in the tumour compared to that of [^111^In]In-DTPA-BV may be explained by [^125^I]I-BV faster metabolism and efflux of the radioactive metabolite from the tumour [208]. Radioconjugate of BV labelled with therapeutic iodine-131, was discussed by Ashrati et al. and Kameswaran et al. [234,235]. [^131^I]I-BV radioconjugate showed high stability, high and specific binding and internalisation rate into SKOV-3 ovarian cancer cell xenogafts in mice model [234]. Kameswaran et al. investigated cell binding studies and biodistribution pattern of [^131^I]I-BV radioconjugate, which were tested in HUVEC, U937 and A375 cell lines and melanoma bearing mice, respectively [235]. Additionally, this radioconjugate was used both as diagnostic and radioimmunotherapy agent due to gamma and beta radiation emitted by theranostic ^131^I radionuclide [234,235]. There are also some reports on BV labelled with I-124 radionuclide [236,237,238], however, these works focus on effective evaluation of VEGF imaging by PET method using different antibodies labelled with I-124 and they will be discussed later.

Radioconjugates of BV and diagnostic copper-64 radionuclide were reported by Paudyal et al., Zhang et al., and Chang et al. [46,47,200,239,240,242]. In the case of BV radiolabelling reactions chelators DOTA [239] or NOTA [200,240] were used. Paudyal et al. using in vivo stable [^64^Cu]Cu-DOTA-BV radioconjugate, demonstrated a strong dependence between tumour size and tumour vasculature expression of VEGF and significant correlation between VEGF levels and accumulation of labelled antibody in human colorectal cancer HT29 mice xenografts. Tested radioconjugate was found to be promising radiopharmaceutical for non-invasive VEGF expression imaging, as well as a powerful tool for patient stratification for their potential BV-based AAT [46,47,239]. Chang et al. evaluated VEGF specific PET radiotracer [^64^Cu]Cu-NOTA-BV applied for imaging VEGF overexpressing renal carcinoma cell line 786-O xenografts (injected subcutaneously into the ears of athymic NCr-nu/nu mice) and for monitoring of tumour response to everolimus inhibition of mTOR kinase [200]. Similar to [^89^Zr]Zr-N-suc-Df-BV [217,218,219], [^64^Cu]Cu-NOTA-BV was considered a novel biomarker for monitoring of the disease status after treatment with rapalog in mTOR kinase inhibitor therapies. Zhang et al. focused on dual-labelled BV ([^64^Cu]Cu-NOTA-BV-800CW), with PET radionuclide copper-64 and NIRF dye 800CW for VEGF imaging of human glioblastoma cells U87MG (overexpressing VEGF-A_121_, VEGF-A_165_ and VEGF-A_189_), where mice model xenografts showed good linear correlation between recorded in vivo PET and NIRF results [47,240]. Such PET/NIRF agents ([^64^Cu]Cu-NOTA-BV-800CW or [^89^Zr]Zr-N-suc-Df-BV-800CW [229,230]) could be utilised in many clinical applications, e.g., disease diagnosis, patient stratification, treatment monitoring, image-guided surgery, etc.

Further reports have studied radiolabelled ramucirumab and ranibizumab, much less commonly used anti-VEGF antibodies in AAT.

Ramucirumab (RamAb, Ram, also known as IMC-1121B, mean serum half-life 14 days), sold under the trade name Cyramza, is a fully human IgG1 class mAb dedicated to treatment of various malignancies, including hepatocellular, colorectal, gastric and lung cancers, as well as a second drug after prior treatment with fluoropyrimidine or platinum-containing chemotherapy.

Ramucirumab labelled with diagnostic radionuclide copper-64 (using NOTA chelator) was described by Luo et al. and Laffon and Marthan [243,244]. Luo et al. presented cell binding studies performed on HCC4006 and A549 cell lines with high and low VEGFR-2 expression, respectively, while biodistribution assays were performed on lung tumour-bearing mice model. The obtained results showed specific binding of [^64^Cu]Cu-NOTA-RamAb to extracellular VEGFR-2 and, moreover, greater selectivity than BV [243]. Laffon and Marthan repeated Luo et al. study and determined three independent kinetic parameters, namely, uptake rate constant, release rate constant and fraction of free tracer in blood and interstitial volume [244]. They showed that the applied evaluation approach may be useful for characterisation of radiotracer and assess patient response to VEGFR-2-targeted therapies.

Deferrioxamine-conjugated Ram labelled with zirconium-89 was synthesised and evaluated by Li et al. [245]. The binding profile of [^89^Zr]Zr-N-suc-Df-Ram radioconjugate was tested using three different prostate cancer cell lines: PC-3, LNCAP and LAPC-4 and showed that radioconjugate conserved Ram affinity and selectivity towards VEGFR-1 and VEGFR-2. PET imaging was also conducted on mice models bearing subcutaneous xenografts of above listed cell lines. The reports suggested the radiocompound could be utilised as a tracer for in vivo VEGFR-2 expression monitoring and patient stratification for AAT.

Janousek et al. described various Ram-based radioconjugates differing in technetium-99m labelling method (direct or indirect method with the use of HYNIC or DTPA bifunctional chelating agents) [246]. In vitro saturation binding studies were performed with overexpressing VEGFR-2 cell lines PC-3 and SKOV-3. All obtained potential radiopharmaceuticals [^99m^Tc]Tc-Ram, [^99m^Tc]Tc-HYNIC-Ram and [^99m^Tc]Tc-DTPA-Ram showed approximately one order of magnitude lower affinity towards the targeted receptor compared to that of the natural antibody, while the binding specificity was conserved. [^99m^Tc]Tc-Ram and [^99m^Tc]Tc-DTPA-Ram exhibited slightly worse binding to VEGFR-2 than [^99m^Tc]Tc-HYNIC-Ram did.

Ranibizumab (Ran, serum half-life 2–6 h) is an antigen-binding fragment (Fab) of BV, which has higher affinity to all soluble and matrix bound human VEGF-A isoforms than origin mAb. Ran is used mainly for macular degeneration treatment.

Nagengast et al. employed Ran labelled with zirconium-89 to evaluate the efficiency of sunitinib cancer treatment [16,247]. VEGF imaging with [^89^Zr]Zr-N-suc-Df-Ran radioconjugate in mice bearing human cancer xenografts was found to be superior compared to imaging using radiotracers [^18^F]FDG and [^15^O]H_2_O. The results of tumour imaging with [^89^Zr]Zr-N-suc-Df-Ran corresponded with tumour growth changes and showed differences in response to AAT among tumour areas. Hence, [^89^Zr]Zr-N-suc-Df-Ran allowed non-invasive dynamic visualisation and quantification of VEGF signalling and could potentially become a biomarker for AAT.

Christoforidis’s research group described the labelling of Ran with iodine-124 and comparison of physicochemical and biological behaviour of [^124^I]I-Ran versus BV- and aflibercept-based ^124^I-radiocompounds in age-related macular degeneration [236,237,238]. Aflibercept belongs to VEGFR-based binding peptides and will be described in more detail in the following section of this article [248]. Experimental PET/CT studies on rabbit model of both [^124^I]I-Ran and [^124^I]I-BV showed no leakage from the vitreous cavity, which indicated the possibility of corresponding therapeutic radiopharmaceutical application in the future [46,236]. The application of [^124^I]I-Ran and [^124^I]I-BV radiocompounds showed shorter intravitreal retention of [^124^I]I-Ran compared to that of [^124^I]I-BV and significant reduction of intravitreal retention after vitrectomy and lensectomy in the case of both radioagents [237]. Christoforidis et al. studied PET/CT biodistribution and pharmacokinetic of [^124^I]I-Ran, [^124^I]I-BV and [^124^I]I-aflibercept in non-human primate model, as well as the potential clinical application of the tested radiocompounds [238].

In summary, it is worth emphasising the frequently reported information that mAb AAT was ineffective in some patients and the results of VEGF imaging levels were not always consistent with the actual amount of this parameter, determined by e.g. ELISA method. A major cause of antibody therapy failure may be due to mAb impossibility to cross the blood brain barrier to reach its target, or that the target is completely unavailable. However, this may be related with tumour size, tumour heterogeneity and patient subjective sensitivity to treatment. Therefore, numerous novel approaches in nuclear medicine insist on personalised treatment, based on previously developed markers that allow prediction of treatment outcome and verification of course of therapy at its every stage. Notably, the most frequent examined radiopreparation, BV labelled with ^89^Zr, was employed in numerous registered clinical trials for imaging of VEGF expression in pulmonary arterial hypertension and various cancers.

Concise information concerning radiolabelled anti-VEGF and anti-VEGFR antibodies are presented in Table 2.

### 3.3. Radiolabelled Peptide-Like Structure Ligands Used for VEGF/VEGFR Imaging

Radiolabelled ligands with peptide-like structure used as radiotracers for VEGF/VEGFR system imaging create a relatively small group, with only few reports describing such compounds [44,238,248,249,250].

Christoforidis et al. conducted iodine-labelled aflibercept studies to evaluate its pharmacokinetic properties by sequential ocular imaging after the intravitreal administration radiocompound [238,248]. Aflibercept is a recombinant fusion protein composed of fragments of human VEGFR-1 and VEGFR-2 extracellular domains combined to Fc fragment of human IgG1. This drug was specific towards age-related macular degeneration disease. Uniquely, aflibercept could bind to both sides of the VEGF dimer, forming an inert 1:1 complex, also called “VEGF trap”. PET/CT imaging study with [^124^I]I-aflibercept in rabbit model showed the radiocompound presents mainly in the vitreous, but unfortunately, the detectable radioactivity was recorded also in the thyroid gland [248]. The comparison of pharmacokinetic properties of [^124^I]I-aflibercept with those of [^124^I]I-Ran and [^124^I]I-BV, already mentioned in the previous section, determined in PET/CT study in owl monkey model, showed that [^124^I]I-aflibercept radiotracer possessed the shortest intravitreal retention time and relatively short and low levels in serum, heart, liver and distal femur [238].

Hao et al. and Cai et al. published simultaneously and discussed the application of peptoids labelled with diagnostic radionuclides for non-invasive VEGFR-2 imaging [249,250]. Peptoids, poly-N-substituted glycine, e.g. GU40C or GU40C4, belong to the peptidomimetics group, in which side chains are attached via nitrogen atoms of the molecule backbone instead of α-carbon atoms as it is in case of peptides, resulting in peptoids peptidase/proteinase resistance [250]. PET imaging study with [^64^Cu]Cu-DOTA-GU40C4 on mice bearing PC3 prostate cancer xenografts revealed a clear and unambiguous radiocompound accumulation in VEGFR-2 positive PC3 tumours [249]. The reports suggested that peptoids labelled with diagnostic or therapeutic radionuclides can play a significant role in cancer diagnosis and treatment.

It is also worth mentioning that research was conducted on the potential application of cyclic nanopeptide RRL containing sequence Cys-Gly-Gly-Arg-Arg-Leu-Gly-Gly-Cys with terminal intramolecular disulphide bridge, for the precise and quantitative characterisation of tumour angiogenesis [44]. The binding specificity of RRL towards tumour-derived endothelial cells and VEGFR-2 protein was confirmed using optical method (FITC-RRL orAlexa 680/800-RRL) and ultrasonic imaging (MB-RRL). After tyrosine attachment to the amino terminal Cys^1^ and then conjugate labelling with iodine-131, the obtained [^131^I]I-Tyr-RRL was stable in saline and human serum. The biodistribution study of [^131^I]I-Tyr-RRL on PC3 xenograft mice showed specific accumulation of radiocompound in the tumour and according the report, [^131^I]I-Tyr-RRL could be considered as a drug for tumour radioimmunotherapy.

Rezazadeh et al. described an innovative approach for preparing peptide-based VEGFRs imaging radioagents [251]. They synthesised and studied the physicochemical and biological properties of technetium-99m labelled _D_(LPR) peptide, which is a retro-inverso peptidomimetic derivative of _L_(RPL) peptide known to have high affinity for VEGFRs [252]. Using HYNIC as a bifunctional chelator and tricine/EDDA as coligands, two radiotracers, [^99m^Tc]Tc-peptide1 and [^99m^Tc]Tc-peptide2 were synthesised, wherein the chelating moiety was attached to either the C-terminus or N-terminus of _D_(LPR) peptide, respectively. The chemical and biological tests performed in terms of their potential application for VEGFR imaging showed that both radiotracers had high stability in saline and human serum and high specific binding to VEGFR-1 and NRP-1. Unfortunately, the accumulation of both radioagents in the tumour was relatively low, nevertheless, in conclusion authors emphasised that the application of retro-inverso peptides for the synthesis of radiopharmaceuticals was stated as very promising. Concise information concerning radiolabelled peptide-like structure ligands used as radiotracers for VEGF/VEGFR system imaging are presented in Table 3.

### 3.4. Radiolabelled Small Molecular Inhibitors of VEGFR Tyrosine Kinase

Radiolabelled small molecular inhibitors of VEGFR tyrosine kinases constitute a relatively large group of radiotracers used for VEGF/VEGFR system imaging [8,16,18,25,26,27,28,40,41,253]. These molecules should exhibit an extremely high affinity towards VEGFR-TKs (Kd in low nanomolar range) and be characterised with logD values from 1 to 3, which allows crossing the cellular membrane by passive diffusion and reaching the intracellular RTK targets [254]. Due to RTKs being overexpressed in many tumour entities, they seem to be a suitable target for cancer imaging. Furthermore, RTK inhibitor labelled with diagnostic radionuclide can be a useful tool for monitoring levels of RTKs in tumour and give valuable information of AAT effectiveness. Currently, about 20 small molecule inhibitors of VEGFR-TKs are approved for clinical use and some of them, labelled with diagnostic radionuclides, have already been used for VEGFR expression imaging.

Multi-targeted RTK inhibitor sunitinib (SU11248) is an anti-angiogenic agent, approved as an anticancer agent in kidney cancer and gastrointestinal stromal tumour therapies. Wang et al. synthesised and described sunitinib labelled with fluorine-18 ([^18^F]5-F-sunitinib, Figure 3) [26,255] and Kuchar et al. reported sunitinib labelled with iodine-125 ([^125^I]5-I-sunitinib, Figure 3) [26,254]. Both radiocompounds were used successfully for PET imaging of angiogenic processes in metastatic renal cell cancer patients.

Sunitinib labelled with technetium-99m was described by Sakr et al. [259]. [^99m^Tc]Tc-sunitinib radiocompound was prepared via direct labelling approach with stannous chloride, reductant of [^99m^Tc]TcO_4_^−^. Unfortunately, a possible structure of the obtained radiopreparation was not presented in the article. Biodistribution study in tumour hypoxia bearing mice showed high target to non-target ratio and rapid organ clearance via urinary and hepatobiliary excretion, indicating that [^99m^Tc]Tc-sunitinib could be considered as a potential selective radiopharmaceutical for tumour hypoxia imaging.

Kniess et al. and Caballero et al. examined the physicochemical and biological properties of methoxy substituted derivative of sunitinib, 5-methoxy-sunitinib and its analogue, [*methoxy*-^11^C]5-methoxy-sunitinib, (Figure 3) [256,257]. The performed molecular dynamic simulation studies showed that fluorine substitution at position 5 of the oxindole scaffold with methoxy group did not affect the orientation of the inhibitor towards VEGFR-2 TK, but influenced their electrostatic and van der Waals interactions. Based on experimental results of the colorimetric MTT assay in two VEGFR expressing cell lines, primary endothelial HAEC and cancer HT29, it was revealed that [*methoxy*-^11^C]5-methoxy-sunitinib was a promising imaging agent of VEGFR-2 TK expressing cells and potential radiotracer for angiogenesis and carcinogenesis evaluation [257].

Another RTK inhibitor with a similar structure to SU11248 is SU5416 therapeutic specifically for metastatic colorectal cancer therapy. Kniess et al. presented the design and labelling of 3-[4′-fluorobenzylidene]indolin-2-one (SU5202, containing the characteristic benzylidene oxindole scaffold existing in SU5416 structure) with fluorine-18 as potential radiotracers for PET imaging [258]. Based on the results of the biodistribution study on mice, an assessment of in vivo metabolic stability in blood and plasma and a small animal PET study on mice with FaDu xenograft, [^18^F]3-[4′-fluorobenzylidene]indolin-2-one ([^18^F]-SU5202) radiocompound (Figure 3) was unstable and, with only moderate IC_50_ affinity to VEGFR-TKs. Hence, it was not a suitable radiotracer for RTK imaging.

Sorafenib (BAY 43-9006) is a part of a group of small molecular inhibitors of the VEGFR-TKs, which is an approved drug for the treatment of primary kidney cancer, advanced primary liver cancer and radioactive iodine resistant advanced thyroid carcinoma. Procedures for the preparation of its analogues labelled with fluorine-18 and iodine-124, dedicated for imaging of diseases accompanied by increased expression of VEGFRs, are the subject of patent reported by Schuller et al. [260]. Examples of the use of these radiocompounds presented in the patent proved that they could successfully act as radiotracers in PET imaging.

Synthesis of radiolabelled sorafenib, containing ^11^C located in the urea carbonyl positions of the molecule ([*carbamate-*^11^C]-sorafenib, Figure 4), was described for the first time by Asakava et al. [26,261]. [*carbamate-*^11^C]-sorafenib imaging on P-glycoprotein and/or breast cancer resistance protein knockout mice using small-animal PET (on P-glycoprotein and/or breast cancer resistance protein knockout mice) with [*carbamate*-^11^C]-sorafenib, revealed the negative influence of P-glycoprotein and breast cancer resistance protein on radioactivity uptake in the brain, resulting in reduced effectiveness of chemotherapy against tumour cells. Poot et al. also prepared and examined [*carbamate*-^11^C]-sorafenib and [*N-methyl*-^11^C]-sorafenib radiotracers (Figure 4) [26,262]. In vivo metabolite studies in rats showed that both radiocompounds were highly stable. Based on PET studies and ex vivo biodistribution experiments in mice carrying RXF393, MDA-MB-231 and FaDu cell line xenografts, it was assessed that [*N-methyl*-^11^C]-sorafenib was more promising target-specific PET tracer and its tumour accumulation could be related to sorafenib treatment response.

Several reports have described various synthetic routes and radiolabelling reactions of multi-target RTK inhibitors, including different diaryl ureas (based on N-phenyl-N′-4-(4-quinolyloxy)-phenyl-urea skeleton with related structure to sorafenib) with PET isotopes fluorine-18 ([^18^F]F-diaryl urea, Figure 4) and carbon-11 ([*carbamate*-^11^C]-diaryl urea, Figure 4) [26,263,264,265]. Both [^18^F]F-diaryl urea and [*carbamate*-^11^C]-diaryl urea radiotracers, containing positron emitter in aryl-ureas moieties, were found to be highly stable in human blood and suitable for utilisation as PET imaging agents for angiogenesis.

Another representative of the small molecular inhibitors acting on the intracellular RTK domain of VEGFR is vandetanib (ZD6474), which belongs to the 6,7-alkoxyanilinoquinazoline family, capable of inhibiting neovascularisation induced by a wide variety of cancers including lung, melanoma, prostate, breast and ovarian cancer. PAQ, an analogue of vandetanib, based on 3-piperidinylethoxy-anilinoquinazoline, exhibited lower IC_50_ value and better binding selectivity to VEGFR-2 compared to VEGFR-1. Synthesis of PAQ and its labelling with positron emitter carbon-11, as well pharmacokinetic behaviour of the obtained radiotracer [*N-methyl-*^11^C]-PAQ (Figure 5) were described by Samén et al. [43]. Biodistribution study in tumour-bearing mice (induced by subcutaneous injection of human breast cancer cell line TUBO or murine melanoma cell line B16F10) demonstrated a heterogeneous uptake of [*N-methyl-*^11^C]-PAQ radiotracer in different tumour models that correlated with the expression of VEGFR-2 determined ex vivo by immunohistochemical analysis. It was concluded that [*N-methyl-*^11^C]-PAQ radiotracer could be used as a PET agent for angiogenic processes monitoring.

Vandetanib and its chlorine analogue chloro-vandetanib are known as potent and selective VEGFR-RTKs inhibitors with low nanomolar IC_50_ values. Gao et al. designed a synthetic routes for the preparation vandetanib and chloro-vandetanib radiolabelled with carbon-11 radionuclide located at nitrogen and oxygen positions in both molecules ([*N-methyl-*^11^C]vandetanib, [*N-methyl-*^11^C]chloro-vandetanib and [*O-methyl-*^11^C]vandetanib, [*O-methyl-*^11^C]chloro-vandetanib, Figure 5) [16,26,266]. These new potential PET agents were obtained with high radiochemical purity, chemical purity and specific activity (370-555 GBq/µmol). The report highlighted the use of synthesised radiotracers for further research as radioagents for VEGFR imaging in tumours and monitoring the therapeutic effectiveness of vandetanib and chlorovandetanib therapy.

Dischino et al. synthesised and studied brivanib (BMS-540215, a pyrrolotriazine-based inhibitor of VEGFR-TKs with potential anticancer activity, specific for treatment of solid tumours, hepato-cellular carcinoma and metastatic colorectal cancer) labelled with fluorine-18 [26,267]. However, the report lacked sufficient information regarding the physicochemical and biological properties of this radiocompound (Figure 6). [^18^F]F-brivanib was obtained with relatively low radiochemical yield and specific activity of 85.1 GBq/µmol. Additionally, biodistribution studies on rodents were performed, however, experimental results were not published. Based on the data, it was not possible to evaluate the usefulness of [^18^F]F-brivanib as PET imaging agent for angiogenic processes monitoring.

Ilovich et al. investigated another potential PET radiotracer, [*methoxy-*^11^C-](trimethoxy-phenyl)-4-indolyl maleimide (Figure 6), with high affinity and selectivity towards VEGFR-TKs, dedicated for visualisation of angiogenic processes [268]. They developed methods for the synthesis of (trimethoxy-phenyl)-indolyl-maleimide and its three derivatives with different halogens at 5-indole position, which in the future could be applied for syntheses of radiotracers containing different PET radioisotopes [268]. Radiolabelling reaction with carbon-11 for the non-halogenated (trimethoxy-phenyl)-indolyl-maleimide compound was performed. The obtained radiopreparation [*methoxy-*^11^C-](trimethoxy-phenyl)-indolyl-maleimide was characterised with desired logD value (1.99 ± 0.04), high stability in human blood and specific binding in cells overexpressing VEGFR-2. However, biodistribution studies conducted in tumour-bearing mice showed unsatisfactory accumulation and low retention of radiotracer in target tissues. In addition, incubation of tested radiotracer with hepatic microsomes displayed very low metabolic stability, which actually disqualified this radiotracer as an agent for non-invasive monitoring of angiogenic processes [26,268]. It was proposed that the maleimide moiety, despite its well-known high affinity and selectivity towards VEGFR-RTKs, is not a suitable linking group.

Anthranilate derivatives are another group of compounds showing antitumour activity, among which 2-([pyridin-4-ylmethyl]amino)-*N*-(3-[trifluoromethyl]phenyl) benzamide (AAL-993) is an anticancer drug targeting VEGFR-RTKs. Hirata et al. designed and synthesised four radioiodinated anthranilate-based radiotracers with related structure to AAL-993 drug [269]. Iodine-125 was located at *m*- and *p*-positions of the phenoxy ring (radiotracers [^125^I]*m*-I-NPAE and [^125^I]*p*-I-NPAE) or phenylamino ring (radiotracers [^125^I]*m*-I-NPAM and [^125^I]*p*-I-NPAM). Incubation of obtained radiotracers in neutral phosphate buffer showed high stability of [^125^I]*m*-I-NPAM and [^125^I]*p*-I-NPAM radiocompounds and rapid degradation of [^125^I]*m*-I-NPAE and [^125^I]*p*-I-NPAE. Therefore, [^125^I]*m*-I-NPAM and [^125^I]*p*-I-NPAM (Figure 7) were selected for further in vivo research. Biodistribution study in human prostate tumour-bearing mice showed high tumour uptake of both radiotracers; however, [^125^I]*p*-I-NPAM exhibited superior tumour targeting and tumour to blood radioactivity ratio in comparison to [^125^I]*m*-I-NPAM. Radiotracer [^125^I]*p*-I-NPAM was recommended for in vivo VEGFR imaging using SPECT/CT method.

A novel approach for the design of new radiotracers for non-invasive visualisation of tumour angiogenesis was described by Mitran et al. [270]. A new conjugate was prepared containing two anti-VEGFR-2 affibody molecules (Z_VEGFR2_ and Bp_2_, characterised with affinities towards VEGFR-2 in the nanomolar range) targeting non overlapping epitopes (so-called biparatopic dimer) of VEGFR-2. The biparatopic affibody dimer called Z_VEGFR2_-Bp_2_ was then labelled with indium-111 using a macrocyclic chelator NODAGA. Obtained [^111^In]In-NODAGA-Z_VEGFR2_-Bp_2_ radiotracer showed excellent stability in PBS and EDTA excess. Targeting specificity, activity biodistribution and imaging properties of [^111^In]In-NODAGA-Z_VEGFR2_-Bp_2_ studied in mice bearing VEGFR-2-expressing xenografts showed that the affinity of radiotracer based on biparatopic affibody conjugate for binding to VEGFR-2 was two orders of magnitude higher than that of individual affibodies. Hence, [^111^In]In-NODAGA-Z_VEGFR2_-Bp_2_ could be used for non-invasive visualisation of the tumour angiogenesis in the preclinical diagnosis of glioblastoma, for therapeutic design and therapy monitoring.

Summary of the above cited articles regarding VEGF/VEGFR system imaging radiotracers reveals that radiopreparations based on small molecular inhibitors of VEGFR-TKs could play a pivotal role in AATs. They allow for referring patients to appropriate therapies, planning the course of these therapies and monitoring the effects of therapies at each stage of treatment. So far, a number of such radiopreparations have been designed and tested and some have proved to be very promising, but none of them have been studied in clinical trials. The search for such radiotracers still remains an ongoing challenge. Concise information concerning radiolabelled small molecular inhibitors of VEGFR tyrosine kinase are presented in Table 4.

### 3.5. Radiolabelled Peptide-Like Ligands for NRP-1 Imaging

Radiolabelled NRP-1 ligands discussed in this chapter can be generally divided into two groups: inhibitors of VEGF-A_165_/NRP-1 complex and tumour penetrating peptides (TPPs) targeting NRP-1. In general, literature data on radiolabelled peptide-like ligands targeting NRP-1 co-receptor is rare. The binding of VEGF-A_165_ to NRP-1 occurs through the b1 subdomain of NRP-1 binding pocket and VEGF-A_165_ fragment of the sequence Cys-Asp-Lys-Pro-Arg-Arg-COOH with C-terminal arginine residue. Therefore, novel NRP-1 ligands can be designed with compatible arginine or lysine located at C-terminus of the R/KXXR/K sequence (R: arginine, K: lysine, X: any amino acids), which has been called “C-end-Rule” [271,272,273]. Moreover, compounds containing such sequence are in vivo competitive with NRP-1 ligands, so they can prevent interaction between VEGF-A_165_ and NRP-1 [274,275].

The only radiocompounds described in the literature, related to the first group, are based on ATWLPPR (A7R) peptidomimetic used alone or conjugated with RGD peptide. A7R is known as in vivo anti-angiogenic and anti-tumour agent [35,36,276].

The second group of radioligands targeting co-receptor NRP-1 includes radiolabelled TPPs, with R/KXXR/K motif [37,38] required for specific NRP-1 targeting and/or internalisation into NRP-1 overexpressing tumour cells. Depending on the position of the R/KXXR/K motif, these compounds can bind to NRP-1 either directly (exposed motif) or indirectly (cryptic motif). In general, TPPs binding to NRP-1 occurs in three steps: (1) TPPs bind to their specific receptors on endothelium; (2) proteolytic cleavage occurs and R/KXXR/K motif is revealed; (3) truncated peptide loses its affinity for its main receptor and binds to NRP-1 [37,38,277] (Figure 8).

There are only few reports in the literature that examine radiocompounds based on A7R labelled with diagnostic radionuclides, namely, technetium-99m for SPECT or fluorine-18 for PET [278,279,280,281,282,283].

Perret et al. investigated the binding affinity of radiolabelled inhibitor of the VEGF-A_165_/NRP-1 complex, [^99m^Tc]Tc-MA-A7R, towards NRP-1 and NRP-2 [278]. The results of the performed research confirmed that [^99m^Tc]Tc-MA-A7R binds to NRP-1 (with no binding to NRP-2). However, biodistribution study on tumour bearing rodents showed low [^99m^Tc]Tc-MA-A7R accumulation in the NRP-1-expressing tumours. Lan et al. studied another radiocompound based on A7R peptide and containing diagnostic radionuclide Tc-99m, [^99m^Tc]Tc-HYNIC-A7R, which was successfully applied for NRP-1 positive tumours imaging on human breast cancer bearing mice model [279].

Wu et al. tested dual-peptide based radiocompound [^18^F]F-Al-NOTA-RGD-A7R for micro-PET/CT imaging of α_v_β_3_ and NRP-1 positive glioma tumours [280]. The performed studies showed higher tumour uptake of this radiotracer compared to those based on single peptides, [^18^F]F-Al-NOTA-A7R and [^18^F]F-Al-NOTA-RGD used separately. [^18^F]F-Al-NOTA-RGD-A7R radiotracer was also employed by Liang et al. for PET/CT imaging in U87MG glioma tumours model [281]. Obtained results showed high specific radiotracer uptake in the tumour; however, similar to previous studies, a noticeable uptake in the kidneys was also observed.

Yufei et al. presented similar dual-peptide radiotracer, [^18^F]F-benzoate-RGD-A7R, for imaging of U87MG tumour model [282]. Cellular binding affinities of benzoate-RGD-A7R were determined through two independent competitive binding assays using [^125^I]I-A7RY or [^125^I]I-RGDyK. MicroPET and biodistribution studies demonstrated clearly visible [^18^F]F-benzoate-RGD-A7R tumour uptake, but unfortunately, also pronounced uptake in kidney, stomach and intestines.

Nanoparticular radiotracer [^18^F]F-*n*-BSA-RGD-A7R based on RGD-A7R hybrid peptide conjugated with modified bovine serum albumin (BSA) was used for biodistribution studies on U87MG tumour-bearing mice [283]. This radiotracer was characterised with enhanced pharmacokinetics properties compared to previously described fluorine radiocompounds [280,281,282]. Experimental results showed higher [^18^F]F-*n*-BSA-RGD-A7R uptake in the tumour and lower in the liver compared to that of [^18^F]F-benzoate-RGD-A7R [281], but still with high kidney uptake.

TPPs found various preclinical applications [38,284,285,286] and some NRP-1-targeting TPPs have also been radiolabelled with diagnostic or therapeutic radionuclides [285,286,287,288,289,290,291,292,293]. These radiopreparations are based on TPPs containing R/KXXR/K motif located at C-terminus (exposed motif): RPAR [37,290], CGNKRTR (tLyp-1) [291], or based on TPPs with R/KXXR/K motif located inside the peptide sequence (cryptic motif): CRGDKGPDC (iRGD) [287,288], CRNGRGPDC (iNGR) [285,289], AKRGARSTA (LinTT1) [39,292] or CLKADKAKC (CK3) [293].

The most investigated TPP radiotracers are radiocompounds based on iRGD peptide. [^68^Ga]Ga-DOTAGA-Ahx-iRGD and [^68^Ga]Ga-NODAGA-Ahx-iRGD were tested in terms of their in vivo imaging capabilities on melanoma murine model [287]. Both radiocompounds showed comparable tumour uptake; however, [^68^Ga]Ga-DOTAGA-Ahx-iRGD exhibited higher accumulation in the kidneys. Carlsen et al. employed the same ^68^Ga labelled iRGD peptide with different PEG linkers in their structure [288]. The biodistribution studies and microPET imaging on mice bearing brain, prostate or melanoma cancers showed successful monitoring of α_v_β_3_ expression and neuropilin 1 related internalisation into the neoplastic cells.

Zhao et al. studied the properties of ^68^Ga-labelled NGR (CD13 ligand) and iNGR (TPP with cryptic motif) peptides conjugated with NOTA ([^68^Ga]Ga-NOTA-NGR and [^68^Ga]Ga-NOTA-iNGR) [289] and DOTA ([^68^Ga]Ga-DOTA-NGR and [^68^Ga]Ga-DOTA-iNGR) [285] for microPET imaging of CD13-positive tumour xenografts. In both studies iNGR based radiotracers revealed higher tumour uptake and longer retention compare to those of NGR-radiotracer, thus, confirming deeper tumoural penetration of iNGR peptides [284].

Adhikari et al. examined a radiocompound containing NRP-1 targeting peptide RPAR, [^99m^Tc]Tc-DO3A-Et-RPAR for possible imaging of NRP-1 overexpressing tumours [290]. Binding assay of NRP-1 overexpressed human glioma cell line, biodistribution and scintigraphy imaging on U87MG xenografts mice showed specific binding to NRP-1 and relatively good tumour uptake, despite the observation of the highest uptake in kidneys and liver, which suggested hepatobiliary excrection.

tLyp-1 peptide coupled with tyrosine at N-terminus followed by iodine-131 labelling, [^131^I]I-Tyr–tLyp-1, was studied on different types of cell lines (NRP-1 positive and negative) and evaluated for biodistribution and SPECT/CT imaging on A549 tumour-bearing mice. The studied radiocompound showed binding to NRP-1 positive cells sensitive to blockage of another TPPs, RPARPAR peptide; however, low tumour and high kidney uptake also indicated dominance of the renal-urinary excretion pathway [291].

LinTT1 conjugated with PEG-PCL polymersomes (PS) labelled with iodine-124 was evaluated for imaging of small triple negative breast tumour in mice, however, mainly in terms of specific targeting to p32 protein [292]. The results of [^124^I]I-Tyr-LinTT1-PS biodistribution and PET-CT imaging showed higher tumour accumulation of LinTT1-polymersomes than peptide-free polymersomes, promising imaging quality and potential application of proposed drug delivery system.

CK3, a NRP-1 targeting peptide with a cryptic KXXK motif [293], was labelled with ^99m^Tc or Cy5 and its binding properties was evaluated towards NRP-1 overexpressed on MDA-MB-231 breast cancer cells. Results of SPECT or NIRF imaging confirmed accumulation of tracers in tumour, but significant high levels were observed also in the kidneys. Due to the presence of a cryptic C-motif, no radiotracer accumulation in the first encountered organs (lung and heart) was monitored in biodistribution studies on normal rabbits. Notably, according to the literature, such compounds cannot be internalised into tumour cells (in contrary to compounds with exposed C-motif) [38,293].

Due to generally low tumour uptake and rapid blood clearance of above mentioned small peptides [290,291,293], Dou et al. examined radiolabelled anti-b1b2 domain of NRP-1 mAb, A6-11-26 [294,295,296] for imaging of NRP-1 positive glioma U87MG tumours [297]. Prepared [^131^I]I-A6-11-26 radiocompound showed high NRP-1 binding affinity in cellular study and poor, but noticeable accumulation in tumours and unfortunately, high accumulation in the kidneys, liver, lungs and blood. The utilisation of antibody fragments or anti-NRP-1 affibody molecules with low-energy emitting radionuclide were proposed for further research.

According to Bumbaca’s opinion [298], poor imaging results of NRP-1 expressing tumours could be caused by the high expression of this receptor in healthy organs and relatively low expression in tumours. Therefore, radiolabelled antibody, [^111^In]In-DOTA-MNRP1685A, was used to evaluate NRP-1 dependent uptake efficiency [298]. The uptake of radiolabelled antibody in tumour and non-tumour tissue was studied on mice bearing human colon carcinoma under simultaneous administration of variable amounts of unlabelled antibody (MNRP1685A) in order to induce receptor saturation in healthy organs. Such action was aimed to receive the most optimal labelled to unlabelled antibody ratio and finally the best tumour visualisation with the lowest uptake in non-tumour tissue. The authors observed MNRP1685A dose-dependent radioactivity increase in tumour and plasma with decrease of a previously high uptake in lungs and liver.

In summary, radiolabelled NRP-1 ligands, possess a dual role—as a vector leading radiotracer to NRP-1 overexpressing tumour cells and anti-angiogenic agent. In our humble opinion NRP-1 receptor abundance on endothelial cells [90,91] and epithelial cells of numerous organs [298] negates to a great extent the application of these radiocompounds. Indeed, high non-target tissue uptakes of radiotracers has been observed. Nevertheless, an advantage of these radiocompounds is the simplicity of the required motif (R/KXXR/K) for NRP-1 binding that allows for numerous modifications within the peptide sequence, which may result in procurement of new radiopreparations with desired properties.

Moreover, such radiocompounds have also an ability to penetrate deep inside the tumour, which could benfit the delivery of therapeutic cargo as radiation source or a cytotoxic drug, equivalently in whole tumour volume. Furthermore, radiocompounds based on more than one peptide, e.g., [^18^F]F-Al-NOTA-RGD-A7R [281], show higher tumour uptake compared to single peptide radiocompounds (synergic effect) and ability to recognise more than one receptor (e.g., α_v_β_3_ integrins and NRP-1), which is their advantage and generally broadens the possibility of their use. Therfore, due to the presented advantages and disadvantages of discussed radiocompounds it seems that only some radiotracers discussed herein (e.g., based on A7R) may be of interest to clinical applications as NRP-1 imaging agents. Concise information concerning NRP-1 targeting radiotracers are presented in Table 5.

## 4. Conclusions

Radiotracers targeting VEGF/VEGFR system presented in this paper represent a diverse group of radiolabelled biovectors based on VEGF ligands and their derivatives, anti-VEGF and anti-VEGFR antibodies, peptide-like structure ligands, small molecular inhibitors of RTK and NRP-1 targeting peptides. Despite the multitude of available tracers, targeting VEGF/VEGFR system is overwhelmingly dominated by radiolabelled VEGF-based radiocompounds and anti-VEGF antibodies. Radiolabelled VEGF-based radiotracers are based mainly on freely diffusible and highly active VEGF-A_121_ isoform, which is medium molecular and hydrophilic peptide showing high in vivo stability and renal clearance. VEGF-A_121_ based radiotracers have high in vitro VEGFRs affinity, while in vivo they struggle to target VEGFRs overexpression, due to their competition with overproduced endogenous VEGF in close tumoural microenvironment.

On the other hand, radiolabelled BV was the most frequently researched mAb among anti-VEGF antibodies. Moreover, in recent years, there have been several clinical trials on efficacy of ^89^Zr or ^111^In labelled BV imaging applied for assessment of treatment effect prediction or as patient stratification method [199,216,218,219,223,299,300]. Although some results of these studies are incomplete and inconclusive, the data obtained from completed trials are a significant development in the pursuit of an effective assessment of the patient treatment outcome and quantitative characteristics of an uptake of evaluated tracer.

Globally looking at above presented research, the vast majority of radiocompounds, although partly referred by the authors of individual publications as promising and worthy of future evaluation, in our humble opinion do not fulfil the requirements of VEGF/VEGFR imaging radioagents due to in vivo instability, target unspecificity or low target affinity. It seems that despite the well-defined knowledge of characteristics of VEGF, VEGFR, and NRP-1 molecular targets and the ability to proper radiotracer preparation, targeting and tumour imaging are still a challenge, due to heterogeneity and changeability of tumour environment [42]. Similarly, results obtained on in vivo models do not transfer directly into results observed in clinical trials. Furthermore, conclusions from the first few clinical trials in this field should be approached with caution. It still needs to be concluded in future research, which of the direct anti-angiogenic approaches is worth being called promising. We already know that it becomes necessary to approach individually to specific groups of patients and not the same therapeutic regimen for each patient. Going one step further, attempts to use a combination of anti-angiogenic and anticancer drugs or a combination of internal radiotherapy with chemotherapy, differing in the mechanism of action, are increasingly considered [301]. The practical benefits of the synergistic approach may prove surprisingly advantageous.

An application of the radiopharmaceuticals offers an exceptional solution for preliminary screening and prediction of patient clinical response before therapeutic intervention. Additionally, radioligand imaging may provide a non-invasive evaluation of the VEGF/VEGFR system targeting and angiogenic processes description, followed by patient stratification for appropriate AAT and final clinical outcome monitoring [130,131,132,133,134]. The multitude of radiopharmaceutical solutions meets the needs of an individual patient and specific treatment regimen. Thereby, targeting VEGF/VEGFR system with radiolabelled tracers is an interesting and prominent approach with clinical benefit that can guide current search for efficient AAT diagnostic methods, patient response evaluation, and further personalised patient treatment approaches.

## Figures and Tables

**Figure 1 cancers-13-01072-f001:**
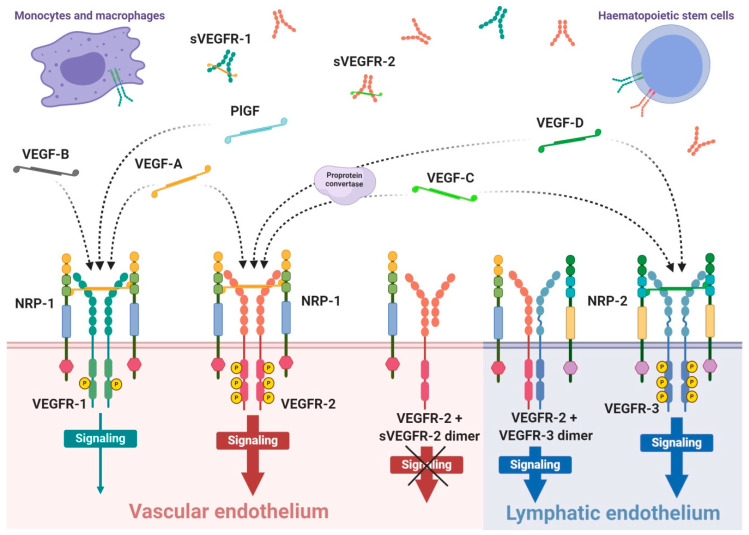
Scheme of expression of VEGF receptors and specificity of VEGF ligands. VEGF receptors occurs mainly as the homodimer transmembrane receptor tyrosine kinases, known as VEGFR-1, VEGFR-2 and VEGFR-3, or in soluble forms defined as sVEGFR-1 or sVEGFR-2. Moreover, surface receptors can create mixed heterodimers or even dimerise with soluble forms. VEGFR-1 expression occurs on vascular endothelium as well as haematopoietic stem cells, macrophages and monocytes. Expression on VEGFR-2 occurs mainly on vascular endothelium, less often on lymphatic endothelium, as well as on the surface of haematopoietic stem cells. The third receptor is mosty expressed on lymphatic endothelium. Conjugation of soluble form with transmembrane receptor preclude VEGF-driven signaling inside the cell. The mammalian VEGF glycoproteins, VEGF-A, VEGF-B, VEGF-C, VEGF-D and PlGF, are expressed as dimers that create different interations with specific VEGFRs, which is indicated by the dashed arrows. Representative VEGF-A glycoprotein binds to VEGFR-1 and VEGFR-2 with significantly higher affinity towards the first receptor. Concomitantly, VEGFR-1 is a specific molecular target for VEGF-B and PlGF, while VEGF-C and VEGF-D selectively bind to VEGFR-3; however, after proteolytic maturation, both VEGF-C and VEGF-D can also bind to VEGFR-2.

**Figure 2 cancers-13-01072-f002:**
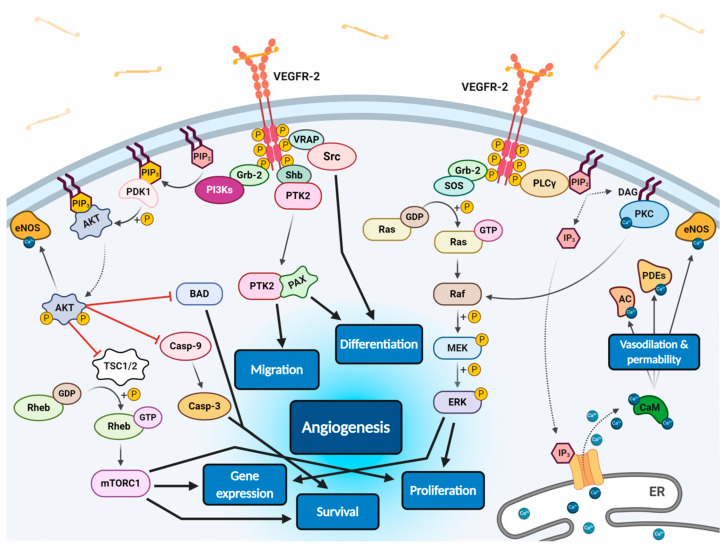
Scheme of endothelial signal transduction of VEGF-VEGFR-2 ligand-receptor molecular complex. The autophosphorylation of receptor tyrosine kinase domains caused by VEGF binding stimulates multiple specific VEGFR-associated proteins (VRAPs) and adaptor molecules inducing concurrent intracellular signalling pathways that promotes proliferation, differentiation, migration, gene expression and apoptosis survival of endothelium leading to angiogenesis.

**Figure 3 cancers-13-01072-f003:**
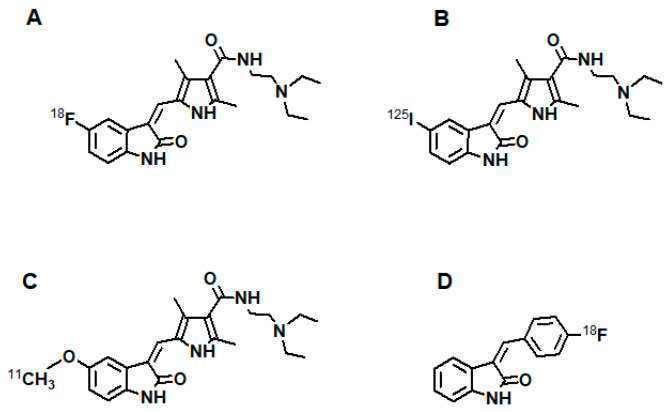
Sunitinib-based VEGFR imaging agents; (**A**): [^18^F]5-F-sunitinib [255]; (**B**): [^125^I]5-I-sunitinib [254]; (**C**): [*methoxy*-^11^C]5-methoxy-sunitinib [256,257]; (**D**): [^18^F]3-(4′-fluorobenzylidene)indolin-2-one [258].

**Figure 4 cancers-13-01072-f004:**
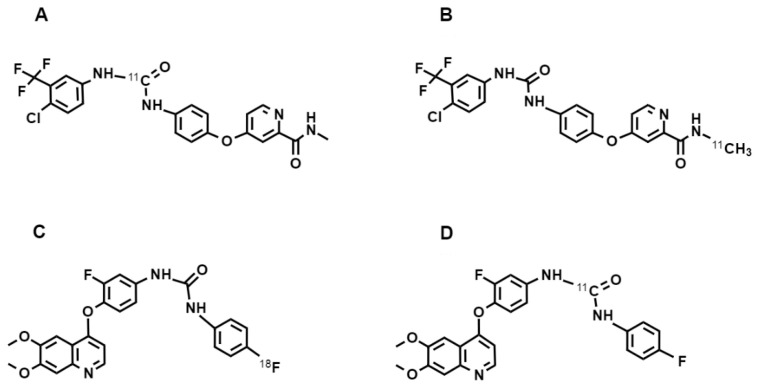
Sorafenib-based and diaryl urea-based VEGFR imaging agents; (**A**): [*carbamate*-^11^C]-sorafenib [261,262]; (**B**): [*N-methyl*-^11^C]-sorafenib [262]; (**C**): [^18^F]F-diaryl urea [263]; (**D**): [*carbamate*-^11^C]-diaryl urea [264].

**Figure 5 cancers-13-01072-f005:**
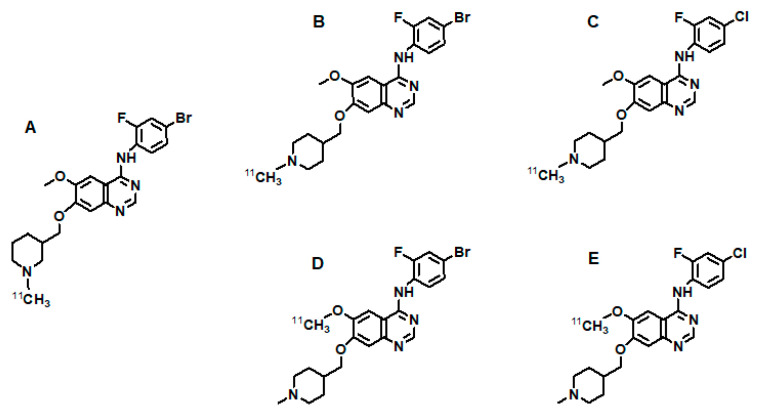
Vandetanib-based VEGFR imaging agents; (**A**): [*N-methyl-*^11^C]-PAQ [43]; (**B**): [*N-methyl-*^11^C]vandetanib [266]; (**C**): [*N-methyl-*^11^C]chloro-vandetanib [266]; (**D**): [*O-methyl-*^11^C]vandetanib [266]; (**E**): [*O-methyl-*^11^C]chloro-vandetanib [266].

**Figure 6 cancers-13-01072-f006:**
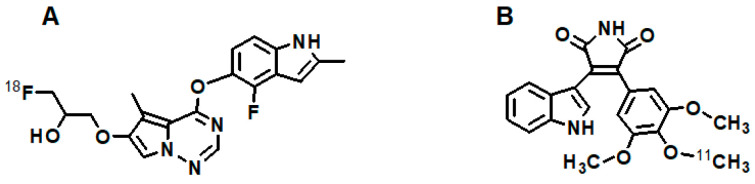
Brivanib-based and diaryl maleimide-based VEGFR imaging agents; (**A**): [^18^F]F-brivanib [267]; (**B**): [*methoxy-*^11^C-](trimethoxy-phenyl)-indolyl-maleimide [268].

**Figure 7 cancers-13-01072-f007:**
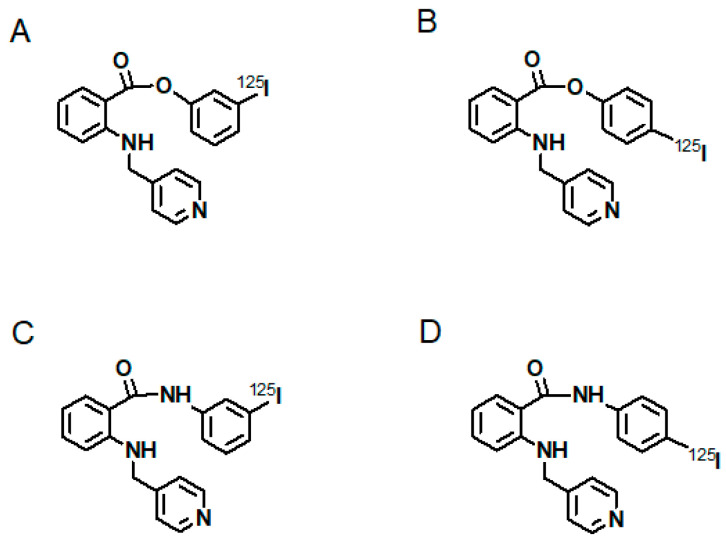
AAL 993-based VEGFR imaging agents; (**A**): [^125^I]*m*-I-NPAE [269]; (**B**): [^125^I]*p*-I-NPAE [269]; (**C**): [^125^I]*m*-I-NPAM [269]; (**D**): [^125^I]*p*-I-NPAM [269].

**Figure 8 cancers-13-01072-f008:**
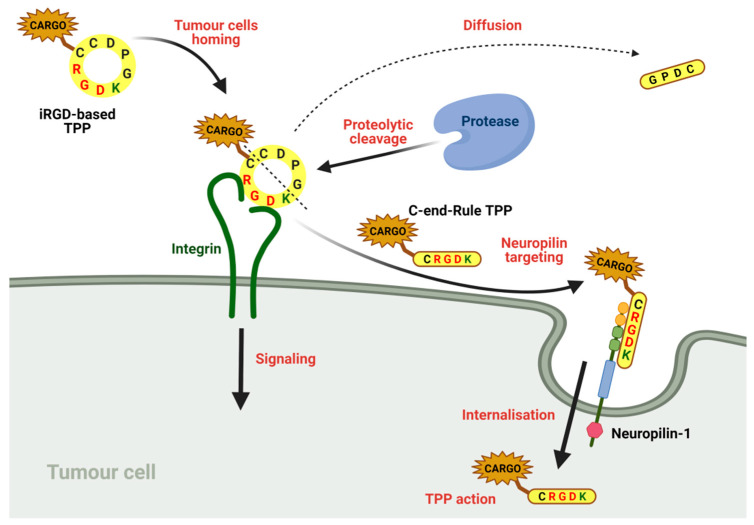
Mechanism of TPP multistep binding and tumour penetration on the example of iRGD peptide.

**Table 1 cancers-13-01072-t001:** Radiolabelled VEGF ligands and their derivatives.

Radiocompound	Aim of Study	References
[^125^I]I-VEGF	identification and characterisation of VEGFR	[144]
[^125^I]I-VEGF-A_121_	investigation of the heparin effect on binding of VEGF-A_121/165_ to VEGFRs; study of the induction of VEGFR-2 tyrosine autophosphorylation by VEGF-A	[146,149]
[^125^I]I-VEGF-A_165_	investigation of the heparin effect on VEGF-A_121/165_ binding to VEGFRs; localisation of VEGFR and quantification of VEGF binding in human kidney; study of VEGF binding to neuropilin-1; study of renal expression of VEGF and VEGFR-2 in experimental diabetes; study on induction of VEGFR-2 tyrosine autophosphorylation by VEGF-A; identification of VEGFR binding sites for VEGF-A_165_	[91,145,146,147,148,149]
[^111^In]In-DTPA-VEGF-A_121_	detection of ischemia related with VEGFRs	[157]
[^64^Cu]Cu-DOTA-VEGF-A_121_	examination of VEGFR expression on different sized human glioblastoma U87MG tumours, in rats myocardial infractions, post-stroke angiogenesis and ischemia; development of VEGFR-2-specific tracer with low renal toxicity	[132,133,150,151,152,153]
[^64^Cu]Cu-DOTA-VEGF_mutant_		
[^64^Cu]Cu-DOTA-VEGF_DEE_		
[^68^Ga]Ga-NOTA-VEGF-A_121_	examination of VEGFR overexpression in U87MG tumour xenograft models	[154,155]
[^68^Ga]Ga-NODAGA-VEGF-A_121_		
[^123^I]I/[^125^I]I-VEGF-A_121_	examination of VEGFR overexpression on various types of human and cancer cells; examination of angiogenesis in LS180 tumour xenograft model	[141,156]
[^123^I]I/[^125^I]I-VEGF-A_165_		
[^99m^Tc]Tc-HuS/Hu-VEGF	examination of VEGFR-2 overexpression in subcutaneous and pulmonary adenocarcinoma tumours	[169]
[^99m^Tc]Tc-HYNIC-C-tagged-VEGF	imaging of tumour vasculature during cyclophosphamide treatment	[159]
[^99m^Tc]Tc-HYNIC-scVEGF	multimodal imaging of VEGFRs; description of site-specific protein modification and labelling; identification of accelerated atherosclerosis in diabetes via VEGFRs imaging	[134,142,160]
[^64^Cu]Cu-DOTA-PEG-scVEGF		
[^99m^Tc]Tc-scVEGF-PEG-DOTA		
[^64^Cu]Cu-DOTA-QD-VEGF	dual-modality optical and PET imaging of VEGFR overexpression on U87MG tumour model	[170,171]
[^64^Cu]Cu-DOTA-(AF)-SAv/biotin-PEG-VEGF-A_121_		
[^99m^Tc]Tc-scVEGF	usefulness of direct labelled VEGF-radiocompound; imaging of VEGFR expression changes in breast cancer xenografts under sunitinib treatment and HT29 xenografts during pazopanib treatment	[139,161,162,163]
[^68^Ga]Ga-HBED-CC-PEG-scVEGF	imaging of VEGFRs in different human xenografts in mice	[164,165]
[^68^Ga]Ga-NOTA/DOTA-PEG-scVEGF		
[^18^F]FBEM-scVEGF	VEGFRs imaging on mouse models with xenografts of various tumour cell lines	[140]
[^89^Zr]Zr-DFO-PEG-scVR1	independent imaging of VEGFR-1 and VEGFR-2 on breast cancer on 4T1luc mice model	[166]
[^89^Zr]Zr-DFO-PEG-scVR2		
[^99m^Tc]Tc-scVR1-PEG-DOTA	selective imaging of VEGFR-1 and VEGFR-2 in atherosclerotic lesions on diabetic and non-diabetic mice	[167]
[^99m^Tc]Tc-scVR2-PEG-DOTA		
[^61^Cu]Cu-NOTA-K3-VEGF-A_121_	PET/CT imaging of VEGFR expression on 4T1 tumour-bearing mice	[168]
[^177^Lu]Lu-DOTA-PEG-scVEGF	synthesis and study of therapeutic radiocompound for targeted systemic radiotherapy on MDA231luc tumour-bearing mice; investigation on effectiveness of combinational therapy with doxorubicin	[136,137]
[^64^Cu]Cu-DOTA-VEGF-A_121_/rGel	determination of anti-angiogenic and anti-tumour effects of a vasculature-targeting fusion toxin on orthotopic glioblastoma mice model by multimodal imaging	[172]
[^123^I]I/[^125^I]I-VEGF-A_165_	examination of tumour localisation in patients with gastrointestinal tumours; evaluation of tumour therapy efficiency on athymic mice models; imaging of highly malignant VEGFR-positive osteosarcoma; investigation of prognostic value of imaging in patients with histologically verified brain tumours	[143,177,178,179,180]
[^99m^Tc]Tc-HYNIC-VEGF-A_165_	evaluation of VEGFR overexpressions on various xenograft tumours in mice	[181]
[^111^In]In-hnTf-VEGF-A_165_	imaging of athymic mice bearing U87MG human glioblastoma xenografts by new human-transferin recombinant protein	[182]
[^99m^Tc]Tc/[^111^In]In-DTPA-VEGF-2K	synthesis of new recombinant proteins and comparison their cytotoxicity on overexpressing VEGFR-1 PAE cells	[183]
[^99m^Tc]Tc/[^111^In]In-DTPA-VEGF-2K-NLS		
[^188^Re]Re-MAG_3_-QKRKRKKSRYKS	evaluation of in vivo distribution and tumour imaging in two groups of human ovarian tumour-bearing mice	[184]
[^99m^Tc]Tc-HYNIC-QKRKRKKSRKKH	synthesis and study of novel small peptides as potential drugs for radioactive diagnosis and therapy in A549 tumour-bearing mice	[138]
[^99m^Tc]Tc-HYNIC-RKRKRKKSRYIVLS		
[^188^Re]Re-EC-QKRKRKKSRKKH		
[^188^Re]Re-EC-RKRKRKKSRYIVLS		

**Table 2 cancers-13-01072-t002:** Radiolabelled anti-VEGF and anti-VEGFR antibodies.

Radiocompound	Aim of Study	References
[^124^I]I-HuMV833	antibody distribution study in patients treated with HuMV833	[188]
[^125^I]I-VG76e	VEGFs level imaging in human fibrosarcoma; patient classification for AAT	[193]
[^125^I]SIB-VG76e		
[^125^I]I-SHPP-VG76e		
[^124^I]I-SHPP-VG76e		
[^99m^Tc]Tc-VG76e	detection and inhibition of human breast adenocarcinoma	[194]
[^153^Sm]Sm-DTPA-VG76e		
[^177^Lu]Lu-DTPA-VG76e		
[^177^Lu]Lu-DOTA-VG76e	synthesis and evaluation of novel potential therapeutic anti-angiogenic radioagents	[195]
[^177^Lu]Lu-DTPA-VG76e		
[^125^I]MBs-I-Bt-Avas12a1	studies of biological aspects of angiogenesis	[196]
[^18^F]MBs-SFB-Avas12a1	VEGFR overexpression and tumour angiogenesis imaging	[197]
[^99m^Tc]Tc-HYNIC-chtiosan-Cy5.5-DC101	ischemia monitoring of umbilical vein endothelial cells studied on mice with surgically induced ischemia	[198]
[^64^Cu]Cu-NOTA-BV	imaging of VEGF overexpressing renal carcinoma; monitoring of tumour response to cancer everolimus treatment	[200]
[^111^In]In-DTPA-BV	assessment of new radiotracers‘ application as markers for non-invasive VEGF imaging in tumour microenvironment	[205]
[^89^Zr]Zr-N-suc-Df-BV		
[^111^In]In-DTPA-BV	research on correlation between radiotracer tumour uptake and level of VEGF-A expression, studied on colon cancer metastases to liver and melanoma lesions treated with BV	[206,207]
[^111^In]In-DTPA-BV	visualisation of VEGF-A for prediction of chemotherapy response and patient classification for anti-VEGF AAT	[208]
[^125^I]I-BV		
[^111^In]In-DTPA-BV	imaging of different VEGF isoforms expression	[209]
	study of radiotracer tumour uptake during sorafenib treatment regading to VEGF expression	[199]
[^111^In]In-CHX-A″-DTPA-BV	scintigraphic imaging of VEGF expression; patient stratification for anti-VEGF AAT	[210]
[^86^Y]Y-CHX-A″-DTPA-BV	application for non-invasive assessment of VEGF-A tumour angiogenesis status; possibility of application as a marker in radioimmunotherapy conducted with the use of therapeutic radioconjugate [^90^Y]Y-CHX-A″-DTPA-BV	[211]
[^177^Lu]Lu-CHX-A″-DTPA-BV	application as therapeutic agent for anti-VEGF AAT	[213]
[^111^In]In-DTPA-BV	VEGF expression imaging and radiotracer application for assessment of response to rapamycin renal and breast cancer treatment	[214]
[^111^In]In-DTPA-Bt-BV	application of avidin chase strategy for [^90^Y]Y-DTPA-Bt-BV therapy	[215]
[^90^Y]Y-DTPA-Bt-BV		
[^89^Zr]Zr-N-suc-Df-BV	in vivo non-invasive visualisation of early changes in VEGF levels during treatment with synthetic inhibitor NVPAUY922	[216]
	application as an early biomarker of everolimus AAT	[217,218,219]
	VEGF expression imaging in primary breast cancer and non-small cell lung cancer; study on correlation between radiotracer uptake and of VEGF-A level	[220,221]
	VEGF-A expression imaging before and during AAT with BV/IFNα and sunitinib	[222,223,224]
[^99m^Tc]Tc-HYNIC-BV	scintigraphic imaging of VEGF levels in tumour	[225,226,227]
[^99m^Tc]Tc(CO)_3_-BV		
[^99m^Tc]Tc-DTPA-BV	evaluation of VEGF binding and application in radioimmunoscintigraphy of various cancers	[228]
[^89^Zr]Zr-N-suc-Df-BV/cetuximab-800CW	early photo- and radio-detection of small, established tumours which cannot be identified by current radiological and nuclear techniques	[229,230]
[^89^Zr]Zr-N-suc-Df-BV/cetuximab		
[^89^Zr]Zr-N-suc-Df-BV	study on VEGFR expression in adult and childhood HGG, including DIPG tumours; identification of heterogeneity of pontine glioma lesions smaller than a centimeter; stratification patients for BV treatment	[231,232,233]
[^131^I]I-BV	tumour-targeting evaluation for cancer imaging and treatment	[234]
	targeting VEGF overexpressing cancers therapy; evaluation of application as potential diagnostic and radioimmunotherapeutic agent	[235]
[^124^I]I-Ran	application in age-related macular degeneration treatment	[236,237,238]
[^124^I]I-BV		
[^64^Cu]Cu-DOTA-BV	study on correlation between VEGF expression and tumour uptake of radiotracer; evaluation of possibility for patient stratification for AAT	[239]
[^64^Cu]Cu-NOTA-BV-800CW	VEGF imaging in human glioblastoma; application in disease diagnosis, patient stratification and treatment monitoring	[240]
[^64^Cu]Cu-NOTA-RamAb	VEGFR-2 binding studies of radiopreparation	[243]
[^89^Zr]Zr-N-suc-Df-Ram	radiotracer application for AAT monitoring and patient stratification for AAT	[245]
[^99m^Tc]Tc-Ram	evaluation of affinity to VEGFR-2 receptor	[246]
[^99m^Tc]Tc-HYNIC-Ram		
[^99m^Tc]Tc-DTPA-Ram		
[^89^Zr]Zr-N-suc-Df-Ran	non-invasive dynamic visualisation and quantification of VEGF signaling; radiotracer application for AAT monitoring	[247]

**Table 3 cancers-13-01072-t003:** Radiolabelled peptide-like structure ligands used as radiotracers for VEGF/VEGFR system imaging.

Radiocompound	Aim of Study	References
[^124^I]I-aflibercept	examination of pharmacokinetic properties of intravitreally applied [^124^I]I-aflibercept in the vitreous cavity	[238]
[^64^Cu]Cu-DOTA-GU40C4	evaluation of diagnostic and therapeutic abilities on VEGFR-2 positive prostate cancer	[249,250]
[^131^I]I-Tyr-RRL	angiogenesis imaging in tumour AAT	[44]
[^99m^Tc]Tc-peptide1	application of retro-inverso peptidomimetic derivatives for synthesis of radiotracers with high affinity towards VEGFR-1 and NRP-1	[251,252]
[^99m^Tc]Tc-peptide2		
([^99m^Tc]Tc-HYNIC-retro-inverso peptidomimetic)		

**Table 4 cancers-13-01072-t004:** Radiolabelled small molecular inhibitors of VEGFR tyrosine kinase.

Radiocompound	Aim of Study	References
[^125^I]5-I-sunitinib	imaging of VEGFR expressing tumours and angiogenic processes	[254]
[^18^F]5-F-sunitinib	RTKs in vivo imaging	[255]
[^99m^Tc]Tc-sunitinib	potential radiopharmaceutical for tumour hypoxia imaging	[259]
[*methoxy-*^11^C]5-methoxy-sunitinib	RTKs in vivo imaging and evaluation of AAT efficiency	[256,257]
[^18^F]3-[4′-fluorobenzylidene]indolin-2-one	RTKs in vivo imaging	[258]
[^18^F]sorafenib	imaging of increased expression of VEGFRs	[260]
[^124^I]I-sorafenib		
[*carbamate-*^11^C]-sorafenib	VEGFR expressing tumours in vivo imaging	[261,262]
[*carbamate-*^11^C]-sorafenib		
[*N-methyl-*^11^C]-sorafenib		
[^18^F]F-diaryl urea	VEGFR-positive tumours in vivo imaging in AAT	[263]
[*carbamate-*^11^C]-diaryl urea		[264]
[*N-methyl-*^11^C]-PAQ	imaging of VEGFR-2 expression fluctuations and angiogenesis	[43]
[*N-methyl-*^11^C]vandetanib	VEGFR imaging and monitoring of effectiveness of vandetanib orchloro-vandetanib therapy	[266]
[*N-methyl-*^11^C]chloro-vandetanib [*O-methyl-*^11^C]vandetanib		
[*O-methyl-*^11^C]chloro-vandetanib		
[^18^F]F-brivanib	VEGFR in vivo imaging and angiogenic processes visualisation	[267]
[*methoxy-*^11^C-](trimethoxy-phenyl)-indolyl-maleimide	VEGFR in vivo imaging and visualisation of angiogenic processes	[268]
[^125^I]*m*-I-NPAE	VEGFR in vivo imaging in AAT	[269]
[^125^I]*p*-I-NPAE		
[^125^I]*m*-I-NPAM		
[^125^I]*p*-I-NPAM		
[^111^In]In-NODAGA-Z_VEGFR2_-Bp_2_	VEGFR-2 expression imaging and visualisation of tumour angiogenesis in GBM	[270]

**Table 5 cancers-13-01072-t005:** NRP-1 targeting radiotracers.

Radiocompound	Aim of Study	References
[^99m^Tc]Tc-MA-A7R	determination of peptide binding to NRP-1 and NRP-2	[278]
[^99m^Tc]Tc-HYNIC-A7R	imaging of NRP-1 positive tumours	[279]
[^18^F]F-Al-NOTA-RGD-A7R	investigation of diagnostic ability on dual α_v_β_3_ and NRP-1 positive glioblastoma tumours	[280,281,282,283]
[^18^F]F-Al-NOTA-A7R		
[^18^F]-benzoate-RGD-A7R		
[^18^F]F-*n*-BSA-RGD-A7R		
[^68^Ga]Ga-DOTAGA-Ahx-iRGD	investigation of diagnostic ability on dual α_v_β_3_ and NRP-1 positive tumours	[287,288]
[^68^Ga]Ga-NODAGA-Ahx-iRGD		
^68^Ga labelled iRGD		
[^68^Ga]Ga-NOTA/DOTA-NGR [^68^Ga]Ga-NOTA/DOTA-iNGR	comparison of diagnostic abilities on CD13-positive tumour xenografts	[285,289]
^99m^Tc and RPAR derivative complex	investigation of diagnostic ability on different NRP-1 positive tumours	[290]
[^131^I]I-Tyr-tLyp-1		[291]
^99m^Tc labelled CK3		[293]
[^124^I]I-Tyr-LinTT1-PS	investigation of diagnostic ability of polymersomes conjugated with LinTT1 peptide on triple negative breast cancer	[292]
[^131^I]I-A6-11-26	investigation of delivery ability on NRP-1 positive tumours	[297]
[^111^In]In-DOTA-MNRP1685A		[298]

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
