# Peer review of "The Role of VEGF Receptors as Molecular Target in Nuclear Medicine for Cancer Diagnosis and Combination Therapy"

_cancers, 2021, doi:10.3390/cancers13051072_

Round 1

Reviewer 1 Report

This review article provides a detailed overview of designed, synthesized, and investigated radiolabeled VEGEF/VEGEFR targeting and imaging. Additionally, the authors have discussed physicochemical properties and their possible application of these inhibitors in combination targeted radionuclide therapy. The review is well-written, extensive, and appropriate references are included.

Author Response

Thank you for your appreciation of our effort.

Reviewer 2 Report

The review by Maslowska et al aims at describing the current knowledge about radiolabeled VEGFR`s ligands and their potential clinical use in combinatorial regimen in cancer treatment.

It was a very pleasure to read a such comprehensive and well-written manuscript, with a such interesting topic. I have just some little points:

  • how the authors discuss the problem described in lines 611-614?
  • despite the manuscript is very well described, it is very long. I suggest replacing the text with summary tables, where possible, such as the paragraph describing the tissue expression of receptors or the functional effects of receptor(s) activation (from line 199 onwards)
  • some little mistakes through the text should be corrected (e.g. in line 284 delete "the"; in vivo and in vitro should appear in italic; references should appeared at the end of each sentence)

Author Response

The review by Maslowska et al aims at describing the current knowledge about radiolabeled VEGFR`s ligands and their potential clinical use in combinatorial regimen in cancer treatment. It was a very pleasure to read a such comprehensive and well-written manuscript, with a such interesting topic.

Thank you for that sincere comment.

I have just some little points:

  • how the authors discuss the problem described in lines 611-614?
    The problem mentioned in lines 611-614 is the most significant issue on anti-angiogenic therapy since this concept has been studied in patients, and the topic itself is extremely important from a practical point of view. In paragraph 2.3 preceding Part 3 of this article, we have provided a brief
    introduction to this issue as an introduction to further reading. In brief, “Although correlation between tumour progression and VEGF-A expression is well established, it does not transfer into intended anti-angiogenic therapeutic effects. The reason for this is indicated in the heterogeneity
    of the same tumour between patients, but also between different tumours in an individual patient, that occurs and changes at different stages of the lesion development.” A great description of this issue is presented in the ref 42, R. Longo & G. Gasparini article. In the 3 part of manuscript we
    added short fragment to describe further this problem and emphasize its importance (lines 920-924). Nevertheless, we provided also summary of that issue in conclusion paragraph.
  • despite the manuscript is very well described, it is very long. I suggest replacing the text with summary tables, where possible, such as the paragraph describing the tissue expression of receptors or the functional effects of receptor(s) activation (from line 199 onwards)
    Indeed, the manuscript is relatively long, but it covers about 300 references, while our goal was to find as many works as possible on the topic of the role of VEGF receptors as molecular target in nuclear medicine. We tried to provide as much information as possible in the most concise way
    possible. In our humble opinion, replacing the text fragments related to description of the receptor expression or the functional effects of receptor activation with tables will not
    significantly reduce the size of the manuscript, but will be much less informative. Nevertheless, the proofreading service helped us to reduce the redundant wordings, making the text shorter and of better quality.
  • some little mistakes through the text should be corrected (e.g. in line 284 delete "the"; in vivo and in vitro should appear in italic; references should appeared at the end of each sentence.
    Indeed, some mistakes occur in the text, therefore the entire manuscript was carefully proofread and spelling errors are corrected. Also text edition is improved following journal
    recommendations.

Reviewer 3 Report

The review was written by the authors based on many high impact factor publications. The authors approached the subject in details, meticulously linking all the information. It is a pity that the authors did not take into account their own achievements on this field.

I suggest an editing improvement in Fig. 8, descriptions are made in small font, which makes it very difficult to read the contained text.

In the introduction, I also missed a broader description of the application of biovectors in nuclear medicine.

I recommend the manuscript for the publication after minor revision.

Author Response

The review was written by the authors based on many high impact factor publications. The authors approached the subject in details, meticulously linking all the information. It is a pity that the authors
did not take into account their own achievements on this field.
Thank you for that comment.

I suggest an editing improvement in Fig. 8, descriptions are made in small font, which makes it very difficult to read the contained text.
That is a significant remark, small descriptions in Fig 8 is enlarged. Moreover whole figure concept is rearranged and adjusted to more realistic scale.

In the introduction, I also missed a broader description of the application of biovectors in nuclear medicine.
We briefly broadened that issue in the introduction paragraph (lines 60-64).

I recommend the manuscript for the publication after minor revision.
Thank you for recommendation. Whole text was proofread by native Englishman and rechecked for spelling errors.

Reviewer 4 Report

The review article presented by Katarzyna MasÅ‚owska and colleagues is focused on overview of until now designed, synthesized and studied radiolabelled vascular endothelial growth factor (VEGF) and vascular endothelial growth factor receptors  (VEGFRs) targeting and imaging agents. It is known that VEGF and VEGFRs play a pivotal role in angiogenesis process. The use of inhibitors of angiogenesis promoting factors were found to be a powerful tool in anti-cancer combination therapeutic strategies. The Authors illustrate and discuss several types of molecules already used in targeted VEGF/VEGFR anti-cancer therapy: human VEGF ligands  themselves or their derivatives, anti-VEGFR monoclonal antibodies, VEGF binding peptides or  small molecular inhibitors of VEGFR tyrosine kinases. These molecules labeled with diagnostic or therapeutic radionuclides can become, respectively, diagnostic or therapeutic receptor radiopharmaceuticals. In targeted anti-angiogenic therapy (AAT), a special role is played by diagnostic radioagents  that allow to determine the location of the emerging tumor, to monitor the course of treatment, to predict the treatment outcomes and, first of all, to refer patients for AAT. The Authors also briefly discuss their physicochemical properties and possible application in combination targeted radionuclide tumor therapy

The Authors conclude that targeting the VEGF/VEGFR system with radiolabelled tracers is an interesting and prominent approach with clinical benefit, that guide current search for efficient AAT diagnostic method, patient response evaluation and further personalized patient treatment approach.

The review is interesting. It include a balanced, comprehensive and critical view of the research area. It is well written and easy to read

Minor points:

  1. –The Authors must carefully review the manuscript. There are a lot of typos errors.
  2. –Lines 236-237: Explain the meaning of Shb, SOS and Grb-2
  3. –Line 284: Perhaps something is missing in this phrase.
  4. –Ref. 252: control the name of the journal. It seems wrong. See doi number.

Author Response

The review article presented by Katarzyna Masłowska and colleagues is focused on overview of until now designed, synthesized and studied radiolabelled vascular endothelial growth factor (VEGF) and vascular endothelial growth factor receptors (VEGFRs) targeting and imaging agents. It is known that VEGF and VEGFRs play a pivotal role in angiogenesis process. The use of inhibitors of angiogenesis promoting factors were found to be a powerful tool in anti-cancer combination therapeutic strategies. The Authors illustrate and discuss several types of molecules already used in targeted VEGF/VEGFR anti-cancer therapy: human VEGF ligands themselves or their derivatives, anti-VEGFR monoclonal antibodies, VEGF binding peptides or small molecular inhibitors of VEGFR tyrosine kinases. These molecules labeled with diagnostic or therapeutic radionuclides can become, respectively, diagnostic or therapeutic receptor radiopharmaceuticals. In targeted anti-angiogenic therapy (AAT), a special role is played by diagnostic radioagents that allow to determine the location of the emerging tumor, to monitor the course of treatment, to predict the treatment outcomes and, first of all, to refer patients for AAT. The Authors also briefly discuss their physicochemical properties and possible application in combination targeted radionuclide tumor therapy The Authors conclude that targeting the VEGF/VEGFR system with radiolabelled tracers is an interesting and prominent approach with clinical benefit, that guide current search for efficient AAT diagnostic method, patient response evaluation and further personalized patient treatment approach. The review is interesting. It include a balanced, comprehensive and critical view of the research area. It is well written and easy to read.
Thank you for these valuable and carefully prepared comments.

Minor points:

  • The Authors must carefully review the manuscript. There are a lot of typos errors.
    Indeed, some mistakes occur in the text, thereby the entire manuscript was carefully proofread by native English and spelling errors are corrected.
  • Lines 236-237: Explain the meaning of Shb, SOS and Grb-2
    Thank you for that remark. These abbreviations are explained.
  • Line 284: Perhaps something is missing in this phrase.
    That is one of the typing mistake. It is now corrected.
  • Ref. 252: control the name of the journal. It seems wrong. See doi number.
    Thank you for that careful remark. Indeed, ref 252 was provided incorrectly. Also, we reviewed whole reference section.

Reviewer 5 Report

This is a very extensive review on radiotargeting of the VEGF(R)-NRP1-complex. The authors have meticuously gathered and evaluated the results of numerous studies. I esteem this effort very much, although I have to admit that as a Basic Researcher not necessarily involved in clinical application, I am sometimes simply overwhelmed by the load of literature data. The authors shortly drew a more comprehensive picture and evaluated the data with respect to the different Parameters, which could be important for such a radiolabeled target. Whereas, the chapter 3 would Benefit from shortening, the chapter 4 should be Extended to asnwer Questions, such:

(1) how does the VEGF(R) complex-targeting moiety (radiolabelled VEGF, complete or Fragment of anti-VEGF)R) antibody, Peptide-mimetic of VEGF, Kinase-targeting compound, NRP1-targeting compoung) effect the efficacy in being concentrated at the Tumor, penetrating barriers (such as the blood-brain barrier).

(2) would a VEGFR-targeting moiety give a better Resolution as compared to an anti-VEGF-antibody, which, like ist target, is soluble and hence diffusible within the tissue?

(3) are there any differences in deriviatives of VEGF-A121 or VEGF-A165, as These two variants are differently tethered in the extracellular Matrix.

(4) how do the different VEGF-targeting compounds differ in their pharmakokinetics and pharmacodynamics? This Information is mentioned for several individual compounds, but Maybe General principles can be concluded and summarized in a table.

This would help and would give this collection of data a memorizable conclusion.

I also would recommend the authors to have the review re-edited by an English-speaking Person, who could help to find the numerous typographic Errors and to rephrase some easily understandable sentences.

Author Response

This is a very extensive review on radiotargeting of the VEGF(R)-NRP1-complex. The authors have meticuously gathered and evaluated the results of numerous studies. I esteem this effort very much, although I have to admit that as a Basic Researcher not necessarily involved in clinical application, I am sometimes simply overwhelmed by the load of literature data. The authors shortly drew a more comprehensive picture and evaluated the data with respect to the different Parameters, which could be important for such a radiolabeled target. Whereas, the chapter 3 would Benefit from shortening, the chapter 4 should be Extended to asnwer Questions, such:

Thank you for your appreciation of our effort. Indeed, the manuscript is relatively long, but we tried to find as many works as possible related to topic of our interest, and to provide as much information as possible in the most concise way possible. In the third chapter we reduced the redundant wordings, making the text of better quality. Furthermore, the fourth chapter has been expanded with relevant information that you have highlighted and a neat conclusion of the whole article.

  • how does the VEGF(R) complex-targeting moiety (radiolabelled VEGF, complete or Fragment of anti-VEGF)R) antibody, Peptide-mimetic of VEGF, Kinase-targeting compound, NRP1-targeting compoung) effect the efficacy in being concentrated at the Tumor, penetrating barriers (such as the blood-brain barrier).
    In general, radiopharmaceutical accumulation in the tumour depends mainly on the biovector and its affinity towards the receptor, while the radiopharmaceutical distribution depends on its size, charge, and significantly on its lipophilicity. Therefore, a biomolecules with high receptor affinity (e.g. mentioned above VEGF, complete or fragment of anti-VEGF(R) antibody, peptide-mimetic of VEGF, Kinase-targeting compound, and NRP1-targeting compound) were selected as the biovectors, which should provide specific, stable and high radiocompound accumulation in tumour. Penetration of the barriers depends on the size of radiopreparation (molecular mass usually below 500 Da), its charge (usually neutral or possibly negative) and its lipophilicity (preferred range between 2.5 and 4). Each newly designed radiopharmaceutical should meet the above listed requirements, and above all, biovector employed in radiopharmaceutical structure should maintain its affinity towards the receptor. Therefore, modification of the biomolecule, allowing for the chelator attachment and radionuclide binding, should be performed apart from the pharmacophore fragment of the biomolecule. Finally, it can be said that in a properly designed receptor radiopharmaceutical, the presence of a chelator-bound radionuclide has negligible effect on its accumulation in tumour and on its blood-tissue or blood-brain barrier crossing. The aim of this publication was to present and describe the research on radiocompounds (containing biomolecules with affinity to VEGF or VEGFRs) in terms of their action as molecular targets in nuclear medicine for cancer diagnosis and combination therapy.
  • would a VEGFR-targeting moiety give a better Resolution as compared to an anti-VEGF-antibody, which, like ist target, is soluble and hence diffusible within the tissue?
    Imaging resolution is a complex issue sensitive to numerous variables. Apart from the technical aspects of imaging, resolution depends mainly on radionuclide radiation properties [please have a look at Second Paragraph “Ideal characteristics of radiopharmaceuticals” in The Handbook of Radiopharmaceuticals A. Owunwanne et al.] as well as biovector target affinity, specificity and clearance [please have a look at Fourth Paragraph “Design of radiopharmaceuticals” in The Handbook of Radiopharmaceuticals A. Owunwanne et al.]. Isotope selection is a well-known matter (open to the experimenter), however, the influence of a particular group of biovectors on in vivo imaging is still poorly researched. The only thing that can be said is that the utility of VEGF targeting (mainly on the example of Bevacizumab) is superior to the concept of VEGFR targeting, as we do not see clinical trials with anti-VEGFR antibodies or other VEGFR-targeting molecules.
  • are there any differences in deriviatives of VEGF-A121 or VEGF-A165, as These two variants are differently tethered in the extracellular Matrix.
    Certainly there are. As we provided in 2.1 paragraph VEGF glycoproteins, VEGF-A121 and VEGFA165 derivatives differ mainly in structural composition, due to alternative VEGFA gene splicing. VEGF-A121 is deprived of exon 6th and 7th of VEGFA gene encoded fragments, while VEGF-A165 contains fragment translated from 7th exon. This slight difference affects glycoproteins’ affinity to extracellular heparin and VEGFR co-receptor neuropilin. So, VEGF-A121 is freely diffusible and highly active isoform, evading neuropilin and heparin sulphate, while VEGF-A165 binds to both, what expands its retention in the extracellular matrix.
  • how do the different VEGF-targeting compounds differ in their pharmakokinetics and pharmacodynamics? This Information is mentioned for several individual compounds, but Maybe General principles can be concluded and summarized in a table.
    As you probably know well, pharmacokinetic and pharmacodynamics characterisation of radiotracer result mainly from features of biovectors. In our review we could distinguish basically three major class of biovectors: small molecular non-peptide compounds, medium molecular
    peptides/peptidomimetics, and large molecular mAbs.
    First class consists of lipophilic VEGFR tyrosine kinase inhibitors able to penetrate cellular membrane. They show relatively rapid biodistribution, high volume of distribution, medium biological half-life and low metabolic rate, which favours elimination with faeces. Second class consists of hydrophilic molecules that has limited biomembrane permeability. They show rapid biodistribution, low volume of distribution, short biological half-life and low metabolic rate, however they follow renal route of elimination. Third class consists of mAbs that are highly hydrophilic and charged proteins. They show low biomembrane permeability, slow biodistribution, low volume of distribution, long biological half-life and low metabolic rate via cellular catabolism or elimination with faeces.
    As you mentioned, we have added several pharmacokinetic descriptions at places, where it was relevant. However, instead of general differences between pharmacokinetic features of molecule classes presented above, we provided proven pharmacokinetic characterisation of specific
    radiotraces, which ware mostly evaluated and might be most interesting for readers.

This would help and would give this collection of data a memorizable conclusion.
Thank you for these valuable remarks. It will certainly help to improve the quality of the article.

I also would recommend the authors to have the review re-edited by an English-speaking Person, who could help to find the numerous typographic Errors and to rephrase some easily understandable sentences.
Indeed, some mistakes occur in the text, therefore the entire manuscript was carefully proofread by trusted native Englishman with adequate academic experience.